# Creating speech zones with self-distributing acoustic swarms

Malek Itani ®[1,3] ✉, Tuochao Chen ®[1,3] ✉, Takuya Yoshioka ®[2] ✉ & Shyamnath Gollakota ®[1] ✉

Imagine being in a crowded room with a cacophony of speakers and having the ability to focus on or remove speech from a specific 2D region. This would require understanding and manipulating an acoustic scene, isolating each speaker, and associating a 2D spatial context with each constituent speech. However, separating speech from a large number of concurrent speakers in a room into individual streams and identifying their precise 2D locations is challenging, even for the human brain. Here, we present the first acoustic swarm that demonstrates cooperative navigation with centimeter-resolution using sound, eliminating the need for cameras or external infrastructure. Our acoustic swarm forms a self-distributing wireless microphone array, which, along with our attention-based neural network framework, lets us separate and localize concurrent human speakers in the 2D space, enabling speech zones. Our evaluations showed that the acoustic swarm could localize and separate 3-5 concurrent speech sources in real-world unseen reverberant environments with median and 90-percentile 2D errors of 15 cm and 50 cm, respectively. Our system enables applications like mute zones (parts of the room where sounds are muted), active zones (regions where sounds are captured), multi-conversation separation and location-aware interaction.

For decades, science fiction books and films have featured the ability to manipulate acoustic scenes. In Dune (1984, 2021) and Get Smart (2008), a cone of silence was used to mute conversations from a specific zone in the room. In Rick and Morty (2019), using futuristic technology, the speech of a character in the background was isolated from a cacophony of sounds to reveal its content. Achieving such feats requires the capability to make sense of acoustic scenes by associating spatial context with each of the constituent sounds. Here, we take a step towards this future by introducing self-distributing acoustic swarms, which can automatically create a wireless microphone array distributed across a large area.

Distributing a large number of wireless microphones and speakers across a room has been a long-standing vision in the acoustic and speech communities[1], since it can enable a range of acoustic capabilities and applications. In contrast to commercial smart speakers and conferencing systems where the microphones are co-located, distributing the microphones across a larger area provides the ability to localize sounds in the 2D space. Further, a distributed microphone array has a larger aperture size and hence can achieve better spatial coverage and/or resolution. Such a distributed wireless microphone system can also allow us to better separate an unknown number of concurrent speakers into individual audio streams, which when coupled with the ability to localize speakers in the 2D space, can help create speech zones (Fig. 1A, Supplementary Movies 1–3). For example, we can separate speech and map colocated speakers to different conversation zones; thus, addressing the problem of group-level multi-conversation separation. We can also use this to create mute/active zones where we suppress/capture speech from specific 2D regions in a room. Finally, this can enable location-based speech interaction for smart home applications, where a speech command

[1]Paul G. Allen School of Computer Science and Engineering, University of Washington, Seattle, WA, USA. [2]Cloud and AI, Microsoft, Redmond, WA, USA. [3]These authors contributed equally: Malek Itani, Tuochao Chen. ✉e-mail: malek@cs.washington.edu; tuochao@cs.washington.edu; tayoshio@microsoft.com; gshyam@cs.washington.edu

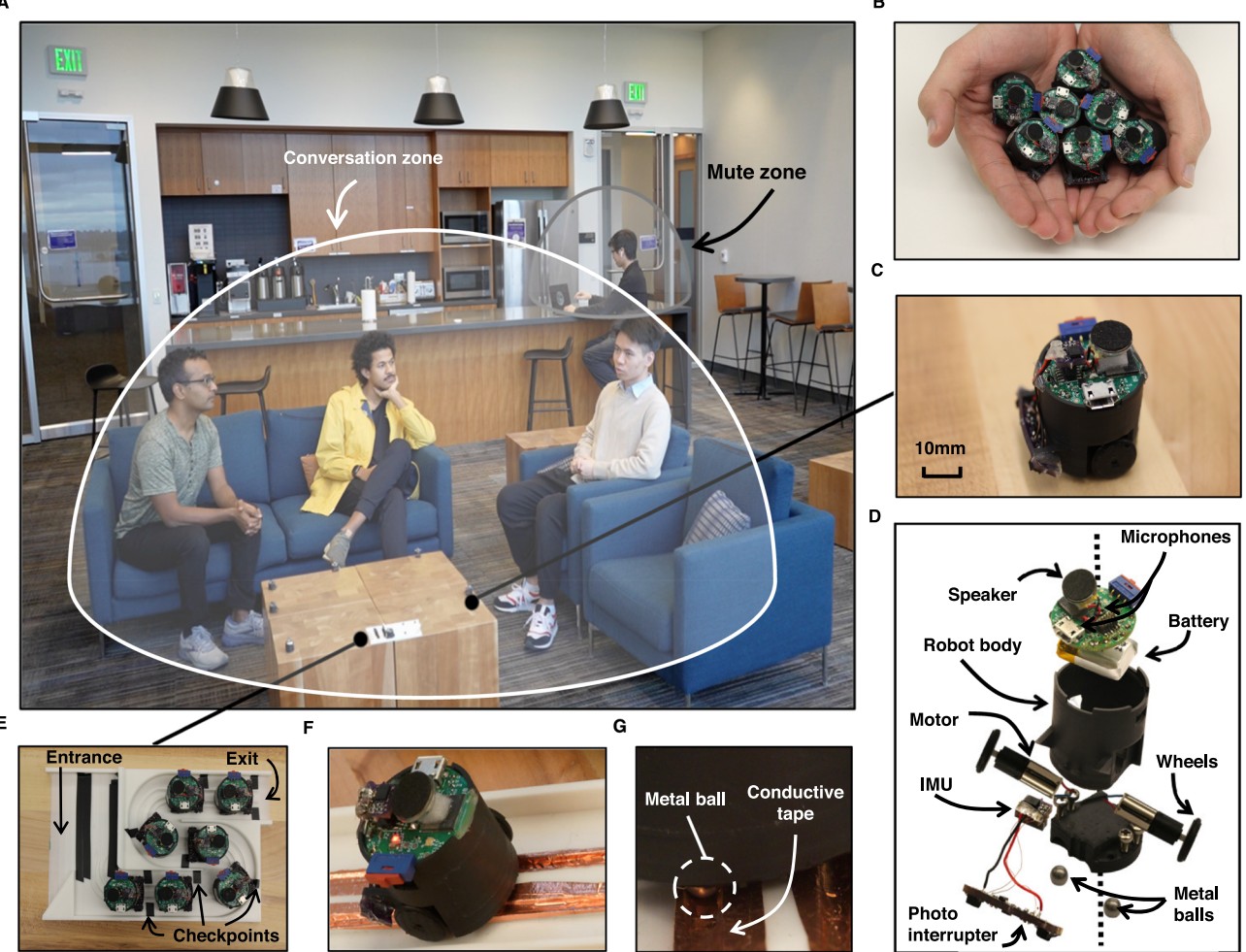

**Fig. 1 | Creating speech zones using our acoustic swarms. A** Shows the acoustic swarm on the table in the front being used to create a conversation zone and a mute zone based on the 2D locations of each of the speakers. **B** Shows a handful of our swarm robots. **C** Shows a closeup with (**D**) showing an exploded view where each robot has a pair of microphones, a speaker, an IMU, two motors, and photo-interrupter to detect surface edges. **E** Shows the base station with an entry ramp, a grooved, meandered track, and an exit ramp. **F** Shows the base with a conductive tape that lines the grooves of the platform. The robot's aluminum balls come in contact with them as shown in (**G**). When powered using a 6 V DC source, this can charge all robots on the platform simultaneously.

could be interpreted differently depending on the speaker's 2D location.

Automating the dispersion of such distributed wireless microphone arrays across a large area is critical for adapting to different environments and spatial constraints, as well as for scaling the system with the number of microphones. Specifically, we need three key capabilities. First, the microphones should be able to disperse themselves across a surface, and adapt to different environments and tasks to efficiently use the available space. Second, given the limited battery-life of wireless microphones, manually dispersing and collecting them for recharging adds to the maintenance cost and is not scalable with the number of microphones. Thus, the microphones should be able to automatically navigate back to a base station to dock and be recharged. Third, to achieve multi-speaker 2D localization, we need the microphones to be wireless synchronized with each other and have the ability to self-localize themselves with centimeter-level accuracy.

We present the first self-distributing wireless microphone array system that can create speech zones. Our work makes three key contributions spanning swarm robotics and deep learning for speech separation and localization: (1) We designed self-distributing acoustic swarms where tiny robots cooperate with each other using acoustic signals to navigate on a 2D surface (e.g., table) with centimeter-level accuracy. We developed navigation techniques for the swarm devices

to spread out across a surface as well as navigate back to the charging station where they can be automatically recharged. Our on-device sensor fusion algorithms combine acoustic chirps and IMU data at the swarm devices to achieve 2D navigation as well as automatic docking at the charging station, without using cameras or external infrastructure. Further, we designed algorithms to ensure that the swarm devices do not fall off the surface and can recover from collisions with other objects. (2) Using the resulting distributed microphone system, we demonstrate the capability to localize and separate speech from different 2D regions. We designed a joint 2D localization and speech separation framework where we use speech separation to achieve multi-source 2D localization while utilizing the 2D location information to improve the speech separation quality. Our architecture has two key components. First, to reduce the search space for 2D localization using neural networks, we run a low-computational complexity signal processing algorithm to prune the search space and then use a speech separation neural network to find the speakers' 2D locations only in the remaining space. Second, in real-world reverberant environments, the speech separation quality can be poor due to residual cross-talk components between speakers. To address this, we incorporated an attention mechanism between speakers by leveraging their estimated 2D locations to jointly compute a much cleaner signal for each speaker and reduce the cross-talk. (3) We demonstrate that our system is

robust to real-world measurement errors while generalizing to unseen environments and running in real-time. We also show proof-of-concept applications like mute zones, active zones, and multi-conversation separation.

## Results

### Self-distributing acoustic swarms

Our goal is to design a miniaturized acoustic swarm that can navigate autonomously on 2D surfaces like tables without cameras or any external infrastructure. Our swarm should meet four key requirements. (1) Since our target application requires maximizing the aperture size, our swarm should spread out to efficiently use the available space. (2) The swarm robots should avoid falling off the surface and also recover from collisions with objects on the surface. (3) Once distributed, the swarm robots should cooperate and accurately localize themselves to compute the absolute 2D position of each robot with centimeter-level accuracy, even in the presence of objects and walls in the vicinity. (4) When the swarm is low on power, it should autonomously navigate back to the base to recharge.

While prior work has presented swarms for miniature robots, none of the existing designs meet the above requirements (see Table 1). In particular, prior work uses external infrastructure to localize swarm robots, including overhead cameras[2], projectors[3], or specialized surfaces[4]. Infrastructure-less cooperative acoustic localization for drones has also been developed in previous works[5,6], however, this focuses on large-scale aerial applications which do not suffer from indoor multipath, do not achieve centimeter-level localization accuracies, and use an array with 4 microphones at each drone to estimate direction of arrival, which cannot fit on our centimeter-scale robots. Infrared sensors[7,8] have been used to estimate the inter-robot distances but this only works for short ranges (<10 cm) and prevents smaller swarms from spreading out very far. Prior work[9,10] also used encoder and IMU data to measure relative changes to a single robot's position, but this approach alone has not been used to achieve the absolute 2D position within the swarm. Finally, while some prior work has demonstrated robot swarms that can self-localize[11–13] and even collaboratively map via SLAM[14] using on-device cameras, these robots are not only large for our applications but cameras also raise a different set of privacy concerns.

**Robot hardware.** Our acoustic robots are based on a Bluetooth low-energy (BLE) module (ISP-1807), which combines a microcontroller (Nordic nRF52840) with capacitors, oscillators, and an antenna (Fig. 1B, C). Each robot is equipped with a gyroscope (STMicroelectronics ISM330DLCTR) and an accelerometer (Memsic MC3419) for odometry, and it is actuated by a pair of micro motors (FA-GM6-3V-25), each driven by a separate motor driver (DRV8837C). Additionally, each robot contains a pair of microphones (TDK Invensense ICS-41352) and a speaker (AS01008MR-3) driven by a digital input Class D amplifier (Maxim Integrated MAX98357A). To detect edges, the robot has a pair of proximity-sensing photointerruptors (GP2S700HCP). The system is powered by a 3.7 V, 100 mAh Lithium Polymer Battery, and a buck

converter (Texas Instruments LM3671) is used to bring the system voltage down to 3.3 V. The battery is charged through a charger IC (Analog Devices LTC4124). To sense battery level, the controller can probe battery information such as cell voltage and state of charge (SOC) through an on-board fuel gauge (Maxim Integrated MAX17048). The main circuit board sits atop a 3D-printed plastic base, which houses the battery and motors (Fig. 1D). The bottom of the robot has a pair of aluminum balls, each connected to the main circuit via a thin wire passing through the robot base. One ball is connected to the system ground, while the other ball is connected to the robot's charger input through a diode. When placed on a pair of conductive rails with a 6 V DC voltage potential, current flows through the balls and wires to charge the robots. Each robot measures 3.0 cm by 2.6 cm by 3.0 cm.

The robots deploy from the plastic base station (Fig. 1E). The station is composed of an entry ramp, a grooved, meandered track, and an exit ramp. Robots can enter the station through the entry ramp. Once inside the base, the robot's aluminum balls slot into the grooved track. This track is used to guide the robots along the base and towards the exit ramp. The sides of the track are lined at specific positions with black checkpoints, which are sensed using the robot's photointerruptors to inform the robot of its position inside the base. Finally, conductive tape can line the grooves of the platform and be powered using a 6 V DC source to charge all robots on the platform simultaneously (Fig. 1F, G).

Our robots wirelessly stream 16-bit audio recordings at 48 kHz via Bluetooth to a host computer for processing to achieve speech separation and localization. Due to the Bluetooth bandwidth limitations of 2 Mbps[15], each robot compresses the recordings in real-time using the Opus Codec. We can simultaneously stream from 7 robots at 48 kHz, without noticeable wireless packet losses, when the audio recordings are compressed down to 32 kbps (Supplementary Fig. 1).

**Acoustic swarm localization.** We use acoustic signals to achieve swarm localization (Fig. 2A, B). The basic idea is to (1) transmit acoustic chirps sampled at 62.5 kHz (see methods), to measure the pairwise distances between robots, and (2) apply a 2D-localization algorithm to estimate the robot's coordinates from the pairwise 1D distances.

To compute the relative 1D distances to all other robots, the robot broadcasts an acoustic chirp. The other robots measure the time-of-flight $\Delta t$ which is converted into the relative 1D distance as $c\Delta t$, where $c$ is the speed of sound[16,17]. A common reference clock is required to compute the time of flight for which we implemented a global clock synchronization algorithm (see supplementary text). The synchronization error never exceeds 1 sample at 62.5 kHz or around 16 μs (Fig. 2C). To find the exact arrival time of a chirp in the presence of an indoor multi-path, we design a dual-microphone algorithm that runs on each robot to combine the chirps received across two microphones on a single robot (see methods). We measured the 1D localization errors in three different scenarios shown in Fig. 2D–F with nearby objects and walls. The 1D ranging errors for the empty desk and the desk near the walls are similar with median errors of 0.48 and 0.45 cm and 90% errors of 1.2 cm and 1.1 cm, respectively (Fig. 2G). Note that

**Table 1 | Comparison with previous centimeter-scale swarm platforms**

| Centimeter-scale swarm platform | Custom infrastructure | sub-100us time sync. | Localization range | Robot size (cm) | Edge detection |
|---|---|---|---|---|---|
| MicroMVP[2] | Camera | No | meter-level | 8 × 5 | No |
| Zooids[3] | Light projector | No | meter-level | 2.6 × 2.6 | No |
| Cellulo (2nd Rev.)[4] | Paper microdot pattern | No | – | 7.5 × 7.5 | No |
| WsBot[53] | Camera | No | meter-level | 3.3 × 3.3 | No |
| Kilobots[54] | Overhead controller | No | ~10 cm | 3.3 × 3.3 | No |
| GRITSBot[55, 56] | Camera | No | meter-level | 3 × 3 | No |
| Ours | None | Yes | ~5 m | 3 × 2.6 | Yes |

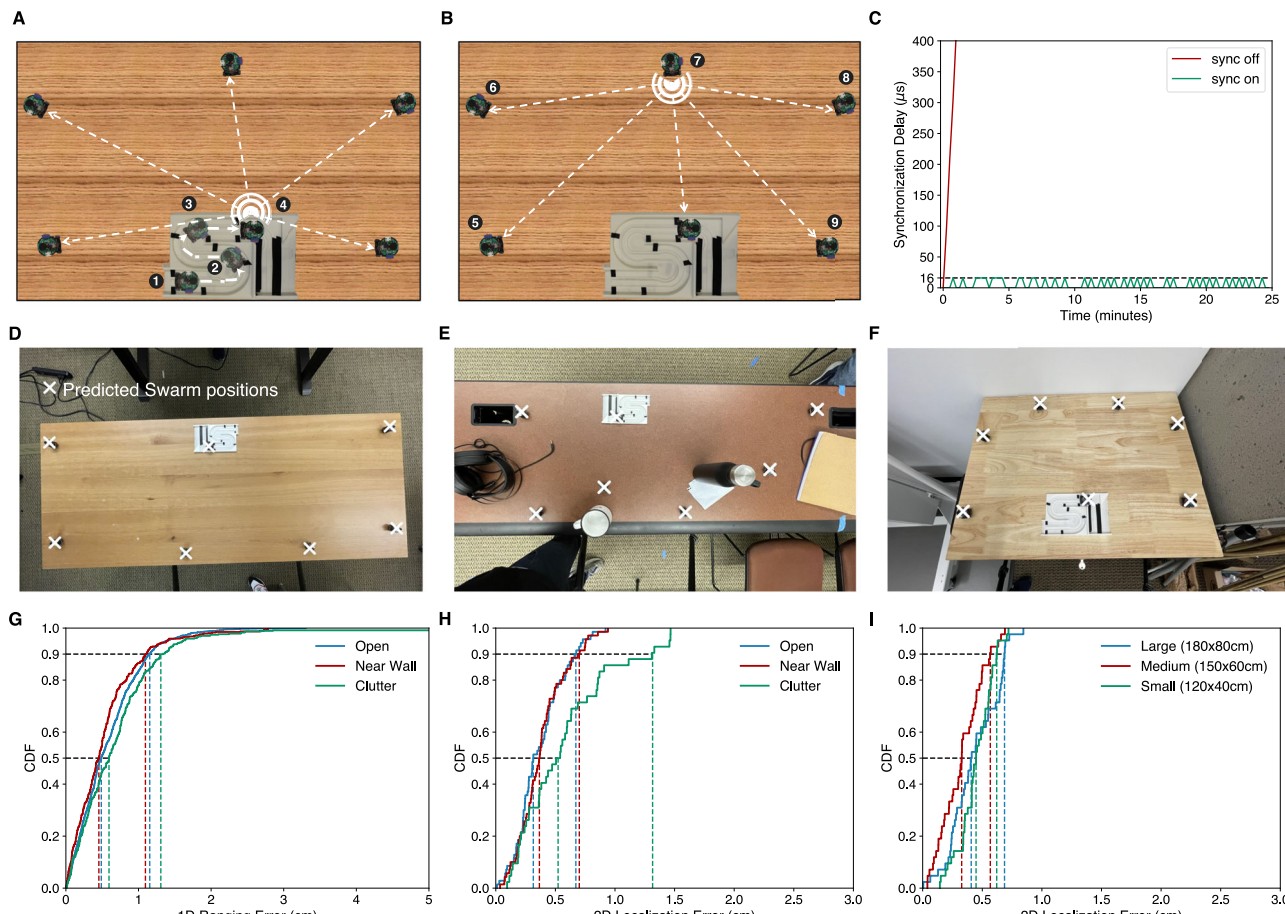

**Fig. 2 | 2D localization of swarm robots. A** One of the robots stays at the platform and moves over all the black checkpoints on the track, emitting acoustic chirps along the way. The remaining robots use these chirps to compute the 1D distances from each checkpoint. **B** The external robots take turns emitting chirps to measure more pairwise 1D distances to improve 2D localization accuracy. **C** Shows the synchronization errors between robot pairs with and without our time synchronization technique. We evaluate our localization algorithm in the scenarios shown in (**D**–**F**) with clutter on the table and walls nearby. **G**, **H** show the 1D and 2D localization errors in these scenarios while (**I**) shows the 2D localization errors for different sized tables. The large, medium, and small table sizes are 1.8 × 0.8 m, 1.5 × 0.6 m, and 1.2 × 0.4 m, respectively.

when the desk had multiple objects, the direct path between some pairs of robots may be blocked, resulting in a long-tailed distribution.

To obtain the absolute 2D coordinates in the base station space and address the long tail issue from 1D localization error, we introduce a pairwise 2D localization pipeline. At the beginning, as shown in Fig. 2A, one of the robots stays at the platform. It goes over all the black checkpoints on the track, emitting acoustic chirps along the way, which the remaining robots use to compute the 1D distances from each checkpoint. This creates virtual landmarks at the checkpoints, which help resolve the rotation and flipping ambiguity during 2D localization (Supplementary Movie 4). Once the robot reaches the last checkpoint, the external robots take turns emitting chirps to measure more pairwise 1D distances to improve 2D localization accuracy (Fig. 2B). Finally, the swarm runs a 2D localization algorithm to estimate the positions of the external robots and compensate for the outliers in 1D localization estimation (see methods). Figure 2H shows that for the empty desk and desk near the walls settings, the median 2D localization errors were 0.37 cm and 0.38 cm, respectively. For the desk with objects setting, the 1D localization long-tail errors were resolved with the 90% 2D error being 1.3 cm. Figure 2I shows that the 2D localization errors across different table sizes were similar, indicating that our localization mechanism can scale to larger surfaces.

**Swarm dispersal.** Consider a swarm of $N > 3$ robots placed in arbitrary order on the grooves of the base station, oriented towards the exit ramp.

Our goal is to disperse the robot swarm as far away as possible across the table while leaving one of the robots at the base. Since the swarms have no prior knowledge of the shape, size, and object occupancy of the desk, we design a heuristic swarm dispersing strategy with two principles: (1) robots expand in equally-partitioned angles, and (2) each robot keeps moving until it either arrives at the desk edge or collides with objects.

Our dispersal mechanism has multiple stages as shown in Fig. 3A–C. The first stage is to sequence and correctly position the robots within the base, before dispersal. Since the robots may be placed in an arbitrary order within the base, the swarm first discovers the robot ordering within the base station. Our intuition is that a forward moving robot first collides with the robot immediately ahead of it. By performing several such collisions between different robots, and using the photointerruptors to identify the base station start and end checkpoints, the robot ordering within the base is obtained (Fig. 3A). The robots then position themselves to be at the base station checkpoints, which they can detect using the photointerruptors. Since the robots may not be evenly distributed along the platform, it is not enough for them to move to the first checkpoint they detect, as two robots may contend for the same checkpoint. Indeed, all robots, except the last robot in sequence, move forward to form a continuous chain starting from the exit ramp. The last robot then moves backward and stops at the end checkpoint. Finally, the other robots, one by one, move backward, collide with the robot behind them, and then move forward to stop at the first checkpoint they detect (Supplementary Movie 5).

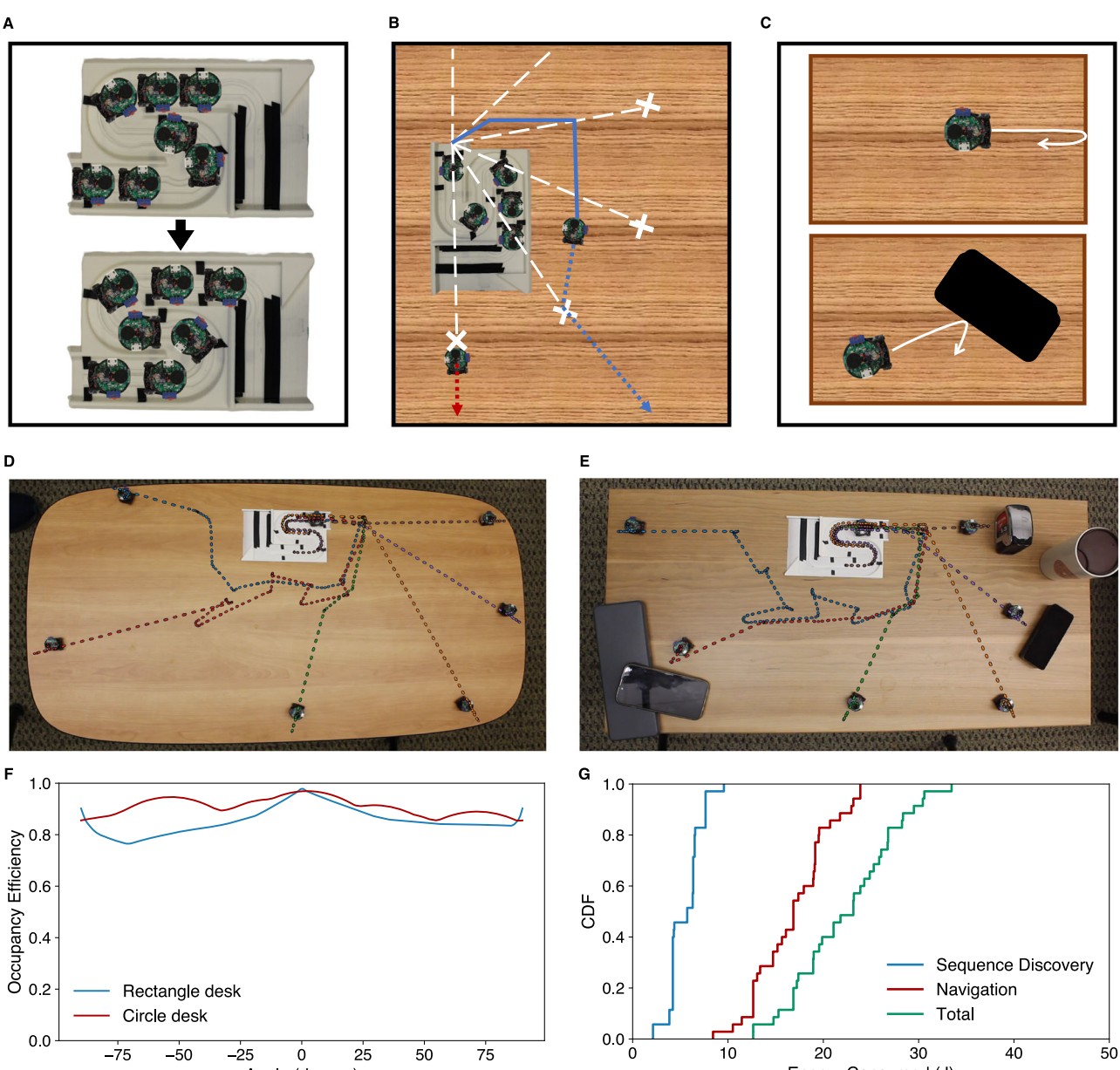

**Fig. 3 | Acoustic swarm dispersal.** Our mechanism for dispersal has three stages. **A** The swarm first discovers the robot ordering within the base station and then correctly positions the robots at the black markers. **B** Next, the robots disperse across equally-partitioned angle. The swarm creates milestones (white crosses) and navigates the robots to the milestones by path-planning using the A* algorithm. The milestones are equidistant from the base and only 4 of the 6 milestones are shown in the figure. **C** The robots use their photointerruptors to detect changes in proximity to the surface and discover edges. They use the IMU to detect collisions. **D**, **E** Shows the paths taken by each robot during swarm dispersal with and without objects on the table. **F** Plots the surface occupancy efficiency as a function of angle for different desk shapes, and (**G**) Shows the cumulative distribution function (CDF) of the energy consumed during sequence discovery and the navigation required for swarm dispersal.

The second stage is to disperse robots across equally-partitioned angles (Fig. 3B). However, the desired directions for the first half of the robots require navigating around the base station and then dispersing at the correct angle. To achieve this, the swarm first creates milestones and navigates the robots to the milestones by path-planning using the A* algorithm. During navigation, each robot tracks its current position, orientation (yaw), and velocity, $(X_t, \theta_t, V_t)$, at each timestamp $t$. The robots use an IMU-based motion model to continuously update their states. To address error accumulation inherent to IMU-based navigation, the swarm uses an IMU and acoustic fusion-based navigation algorithm (see methods). The basic idea is that the remaining robots at the base station cooperate by becoming landmarks for acoustic localization. The moving robot, while in motion, sends acoustic chirps every 200 ms to measure its distance to these landmarks. It then uses these distance estimates to periodically calibrate the IMU-based state and correct the drift during navigation.

Now that the robots are at their designated milestones, they orient themselves to their assigned angles and disperse away from the base station. Since each robot has knowledge about its orientation at its milestone, it can use its gyroscope to rotate to the desired direction and keep moving. The robots expand outwards until they are no longer able to do so. Specifically, they use the photo interrupters to detect changes in proximity to the surface, and they use the IMU to detect collisions (Fig. 3C). Once an edge or collision is detected, the robots back off slightly to avoid falling over edges and stay away from objects (Supplementary Movie 6). Our robots can correctly detect and react to

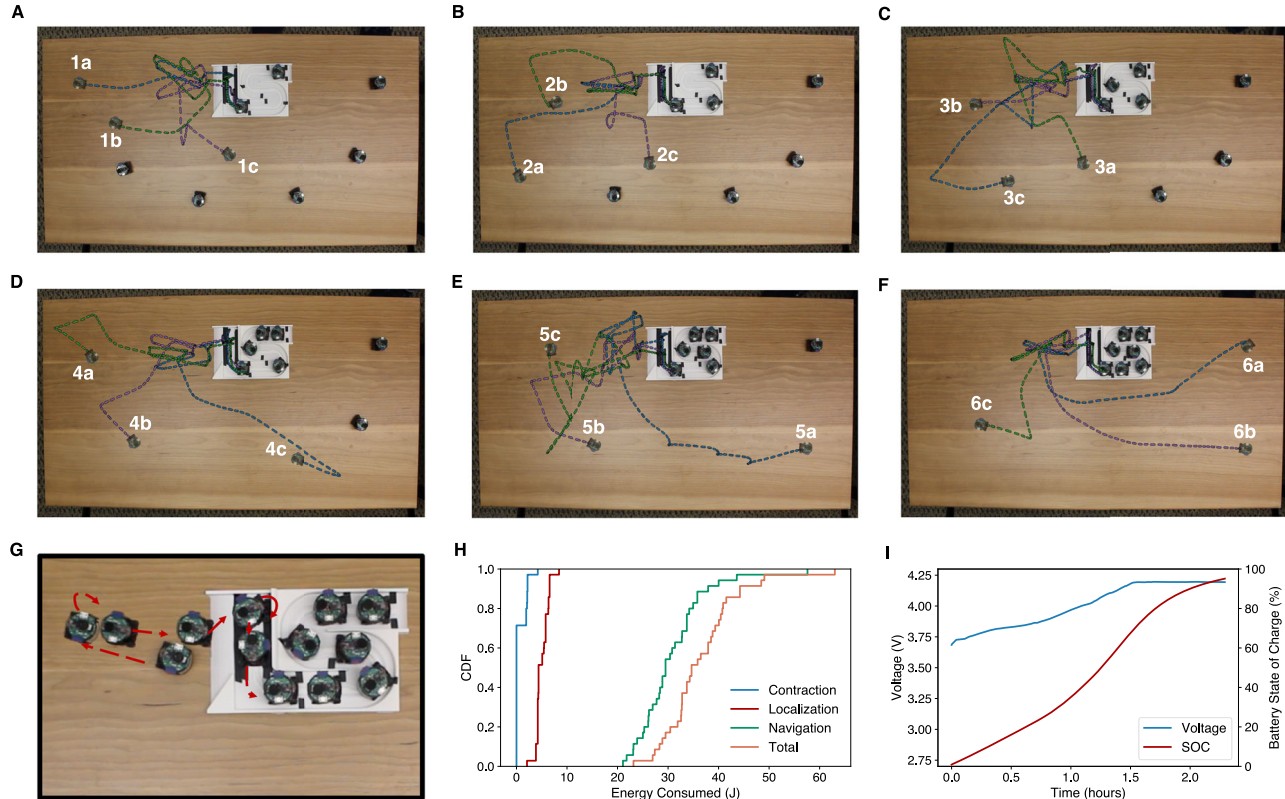

**Fig. 4 | Acoustic swarm returning to the base. A** Shows the paths taken by the first external robot as it navigates its way back to the base from three different initial positions. **B–F** Show the corresponding paths taken by the second, third, fourth, fifth and the sixth external robots, respectively. **G** Shows a timelapse of the maneuvering performed by a robot when it is close to the base to orient itself for docking. **H** Shows a cumulative distribution function (CDF) of the energy consumed for different activities like contraction to the clear zone, localization, and navigation. **I** Shows the voltage and battery state of charge (SOC) as a function of time when the robot is on the charging station.

edges at speeds as high as 18 cm s⁻¹ and detect collisions with objects for speeds as low as 10 cm s⁻¹ (Supplementary Fig. 2). Figure 3D, E show the navigation paths taken by the swarm robots and their behavior near edges and recovery from object collisions. We also characterized the effectiveness of our swarm dispersal using a surface occupancy ratio metric. This is the ratio between the largest possible array size we can achieve for a given table and the actual array size using our distribution strategy. Figure 3F shows that the surface occupancy ratio is above 75% across all directions for different surface geometries. Finally, since the dispersal step is only the initial step for our distributed microphone applications, it should not significantly drain the robots' batteries. Figure 3G shows that the dispersal process consumes on average 22.3 J or 1.7% of the overall battery life.

**Navigating back to the base.** After the designated task is finished or the swarm is low on power, the robots come back to the base station (Supplementary Movie 7). We make two key assumptions: (1) There is a small region, we call a clear zone, around the base that is clear of objects. If the base station is near an edge or wall, this is the region that is part of the surface and is within 25 cm of the base station. This zone can be used for the robots to maneuver and dock with the base station. (2) We can make use of the fact that the robots reached their current positions due to dispersal by moving in a straight path away from the base station. So, we assume that no objects were later placed on this path. As a result, the robots can approach the base station by moving back along that same path. Hence, the robots outside the base station simultaneously approach it. Concurrently, the lone robot in the base station transmits periodic chirps at a rate of 5 Hz that all the other robots use for 1D acoustic ranging to estimate their distances to the base. Using these distance estimates, each of the moving robots comes

to a stop as soon as it arrives within the clear zone. Once all the robots are inside the clear zone, the robots run our 2D localization algorithm to accurately estimate their positions, which are then used to navigate back to the base station. Specifically, the swarm uses our IMU-acoustic navigation algorithm, where the landmarks are the positions of the stationary robots inside and outside the base station. The robots move back to the base station in the same order they exited during dispersal. Once a robot is close enough to the entry ramp, however, its estimated orientation may be erroneous due to gyroscope drifts during navigation. Since a precise orientation estimate is needed to enter the base, the robot executes a simple calibration maneuver to correct the rotation errors. During this time, the robot moves in a straight line and uses several 2D acoustic measurements to estimate its direction of travel, i.e., its orientation. The robot then moves back to the center of the entry ramp. There is a narrow region in front of the base from which the robot can enter, and the robot uses 2D acoustic measurements to verify that it is inside this region. If it is not, it corrects its position by rotating and moving forward in sequence until it is inside the region, after which it enters the base station. It then performs a short sequence of rotations to identify the entry ramp tracks and moves forward to dock with the grooves of the track (see supplementary text). As the robot enters the base, other robots may move to different black checkpoint positions to make room for it inside the base station. The robots in the base accordingly update their positions and are re-used as landmarks for the next robot to navigate back.

Figure 4A–F show the trajectories taken by robots as they move back to the base station. Different plots show trajectories of various stages of the process. At each stage, a single robot moves back to the base station as the remaining robots act as acoustic landmarks. In these figures, we kidnap the moving robot in each figure and place it at

a different starting position and orientation and have it move back to the base station. Figure 4G also shows a timelapse of the maneuvering performed by a robot when it is close to the base to orient itself for docking. Figure 4H, I show the energy consumed during this process and the time it takes for the robot to fully charge, once it is back at the base, which is around 2.5 h.

The time it takes for the swarm to disperse and return to the base is based on two key factors: (1) the size of the table, and (2) the number of robots in the swarm. We recorded the time consumption for the dispersal and return of the acoustic swarm on a 90 cm × 45cm table. For a swarm in the base station to distribute across a surface, the robots need to first identify the order in which they are placed in the base station. For a swarm of 7 robots, this took around 32 s. After this, the robots disperse one by one, taking approximately 11.8 ± 7.0 s per robot dispersal. In total, the sequence discovery and swarm dispersal take around 1 min and 45 s. To return to the base, the robots first moved toward the clear zone in less than 1.5 s. Then, the robot at the base station goes around the landmarks on the base, performing acoustic chirps along the way. This is dependent on the number of landmarks on the base station, and with 7 landmarks, it took around 30 s. Next, the robots outside the base take turns emitting chirps, which took about 3.1 ± 0.02 s per robot. Finally, each robot needs to navigate to the base station one by one, perform the pre-entry calibration maneuver, and successfully dock with the station. This took 40.4 ± 4.5 s per robot.

### Speech separation and 2D localization

We present a new distributed microphone array processing algorithm using our acoustic swarm that performs the following two tasks: (1) localize all speakers in a room without prior knowledge about the number of speakers, (2) separate the individual acoustic signal of each speaker. The algorithm must be robust to microphone position errors and work across different array shapes and sizes even in reverberant real-world environments. While prior work in deep learning proposed speech separation networks[18–21], they did not achieve 2D localization. Recent work also explored distributed microphone arrays[1]. However, they did not satisfy the above goals: they were evaluated in simulated or strongly constrained environments[22–25], required exact microphone positions[26–29], used wired setups to achieve synchronization[26,30,31], localized only 1–2 speakers[31–35], or assumed a priori knowledge about the number of speakers[36–38].

Our algorithm is based on a joint 2D localization and speech separation framework where we use speech separation to achieve multi-source 2D localization of an unknown number of speakers. The computed 2D locations are used to further improve the speech separation performance.

**2D localization via separation.** Let us consider a multi-channel speech separation network that extracts a signal from a speaker if the person's waveforms are found to be aligned across all the microphones while producing a zero signal otherwise. Such a source separation network can be used to examine whether each localized space contains a speaker or not. Specifically, we can align the microphone channels to each location where a speaker may exist by time shifting. To do this, we shift the microphone signals based on the Time Difference of Arrival (TDoA) for each location. The TDoA values are the signal propagation time differences from the candidate location to each pair of microphones. If the location contains a speaker, the shifted speaker signals will be aligned across all the channels while the signals from other locations will be unaligned. Thus, the separation network applied to the time-aligned signals will produce an enhanced speech signal for the target location. Therefore, by checking the output signal amplitude, we can check for the presence of a speaker at each location to count the speakers as well as to obtain their 2D locations.

While our objective is to perform speaker 2D localization, we conduct the search for speakers in the 3D space. This is because the height difference between speakers and microphones introduces additional time delays to the multi-channel signals. To efficiently search for speakers in the 3D space, we combine neural speech separation and a conventional source localization method. Specifically, we first prune the search space using the Steered-Response Power Phase Transform (SRP-PHAT) algorithm[39] (Fig. 5A). SRP-PHAT is a signal processing technique that can achieve coarse localization of the sound source by analyzing the phase differences between all pairs of microphones. SRP-PHAT outputs the power of the signals aligned to each possible candidate point in the search space. We prune the search space by discarding the region with low output power. We then use an attention-based separation model to find the potential speaker locations in the remaining space. The separation model uses a U-Net-style[40] encoder-decoder structure with a transformer encoder[41] bottleneck layer inserted in between (Fig. 5B). This transformer encoder uses self-attention, a mechanism to correlate between different parts of an input sequence when making predictions or encoding information. Here, we use it across the time dimension to encode the relative importance between the utterances at different time instances of the same speaker and get a cleaner output signal. This hybrid approach allows us to avoid searching across the entire 3D space by applying a neural network to every local region, which is computationally very demanding. While SRP-PHAT may not be as effective as deep learning, it can still provide a coarse estimation of the likelihood of speaker presence in a space with much lower computational complexity.

In addition, we utilize the following tricks for robustness and efficiency: (1) Due to indoor reverberation, the direct path and strong reflections of a speaker signal may align at other locations, creating phantom speakers. We eliminate these phantom speakers by clustering similar speaker outputs (see methods). (2) To address the imperfect 2D microphone positions provided by the acoustic swarm, we randomly shift the microphone signals by up to four samples during training. This corresponds to a 2D error of up to 2.8 cm between the swarm robots. (3) To achieve a high spatial resolution without significantly increasing the computational complexity, we first condition the network to search in a larger region with lower resolution around each target location and discard the regions with low output amplitudes. Then, we decompose the remaining regions into smaller sub-regions and run a finer search. Conditioning the network on the large and small regions is done by passing a one-hot vector as a secondary parameter to the network. The one-hot vector is a two-element array with one entry set to zero and the other set to one, depending on whether we are conditioning on small regions or large regions. We provide this vector at each block of the speech separation U-Net (Fig. 5B). (4) For efficient speaker search, instead of discretizing the 3D Euclidean space into uniformly-sized cubes, we divide the 3D space into a set of regions associated with uniformly-spaced discrete TDoA values. The points of 3D space inside each region map to the same discrete TDoA values (see definition of TDoA space in Methods). This also enables generalization to different microphone distributions.

**Separation via 2D localization.** A first-cut solution for obtaining isolated signals of the individual detected speakers is to use the outputs of the separation network employed for the localization. However, in the real-world reverberant environments, the output signal quality of the localization-oriented separation network can be poor due to residual cross-talk components between the speakers. Further, this approach under-utilizes the information we gain from our 2D localization network, i.e., the locations of the other speakers. This information can be leveraged to jointly compute much cleaner signals for all the speakers and reduce the cross-talk. This is achieved by utilizing inter-speaker attention as follows.

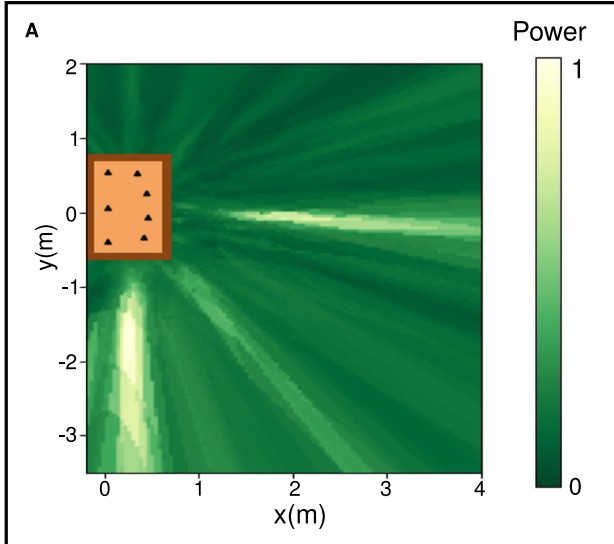

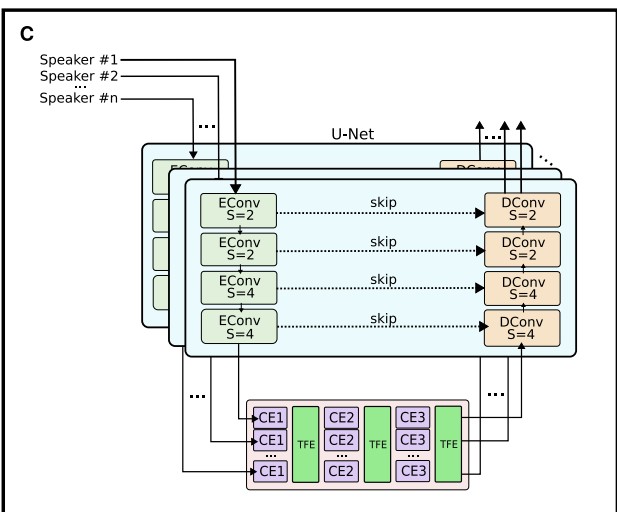

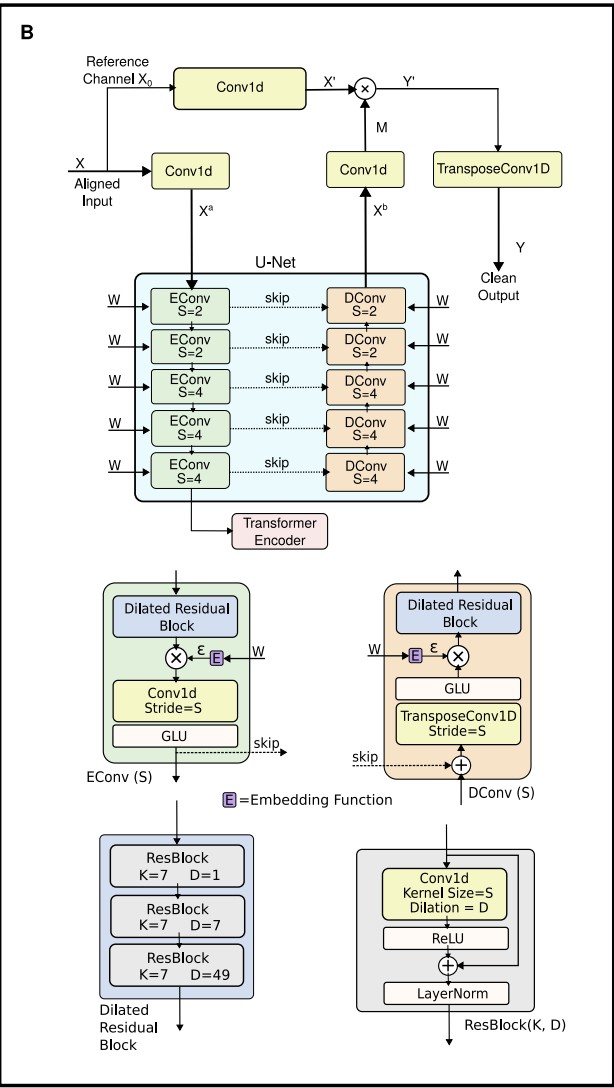

**Fig. 5 | Joint 2D localization and speech separation framework. A** We first run the SRP-PHAT algorithm to prune the search space, and then in (**B**) we use an attention-based separation model to find the potential speaker locations in the remaining space. The separation model is composed of a U-Net encoder-decoder with a transformer encoder bottleneck between them. GLU stands for Gated Linear Unit. **C** shows our network used for speech separation. The encoder and decoder blocks are applied separately to the aligned microphone data for each of the speakers. The bottleneck block first applies temporal self-attention to each speaker individually using a conformer encoder (CE). It then applies self-attention across speakers using a transformer encoder (TFE) to compute attention weights across different speakers. It repeats this multiple times to address cross-talk between speakers.

Since the localization step identifies the number of speakers $S$ and their 2D locations, for each speaker, we align the $M$ microphone signals to their 2D locations and feed the resultant $S \times M$ signals to a new separation model as shown in Fig. 5C. In this model, the encoder and decoder blocks are applied separately to the aligned microphone data for each of the speakers. However, the bottleneck block uses inter-speaker attention to deal with the cross-talk. The bottleneck block first applies self-attention to each speaker using a conformer network[42] along the time dimension, processing each speaker independently (intra-speaker attention). It then uses a transformer encoder that applies attention along the speaker dimension, so that the model correlates information between different speaker channels. The intra- and inter-speaker attention layers are alternately applied to let the network identify and attenuate the cross-talk. Since the inter-speaker attention is performed in the speaker dimension, our architecture can be applied to any number of speakers (see methods).

Figure 6A shows the separation experiment result for an example synthetic mixture of two speech sources, with the corresponding spectrograms depicted in Supplementary Fig. 3. The precision and recall regarding speaker counting were both above 89% even with five concurrent speakers (Fig. 6B). The median and 90-percentile 2D speaker localization errors were 9–10 cm and 32–36 cm for 2–5 concurrent speakers (Fig. 6C, Supplementary Movie 8), respectively. Our algorithm also worked across different microphone array sizes (Supplementary Fig. 4). We evaluate the quality of our separation algorithm using the Scale-Invariant Signal-to-Distortion Ratio (SI-SDR)[43]. In Fig. 6D we show that our technique outperformed the ideal ratio mask (IRM)[44], an oracle speech separation method, by 4.8 dB in terms of SI-SDR improvement (SI-SDRi) over the unprocessed mixture signal for the five-speaker case. Our comparisons with existing transformer (SepFormer[45]) and convolution-based (Conv-TasNet[46]) source separation networks showed improvement across a range of input SI-SDR values for two concurrent speakers, as shown in (Fig. 6E). To create the plots in Fig. 6E, we divide the input SI-SDR into 5000 discrete steps and apply a moving average filter of length 100 on the output SI-SDR. The plot also shows the significant performance contribution of the proposed inter-speaker attention bottleneck block. Figure 6F shows that SRP-PHAT could reduce the search space by a factor of 446 for the two

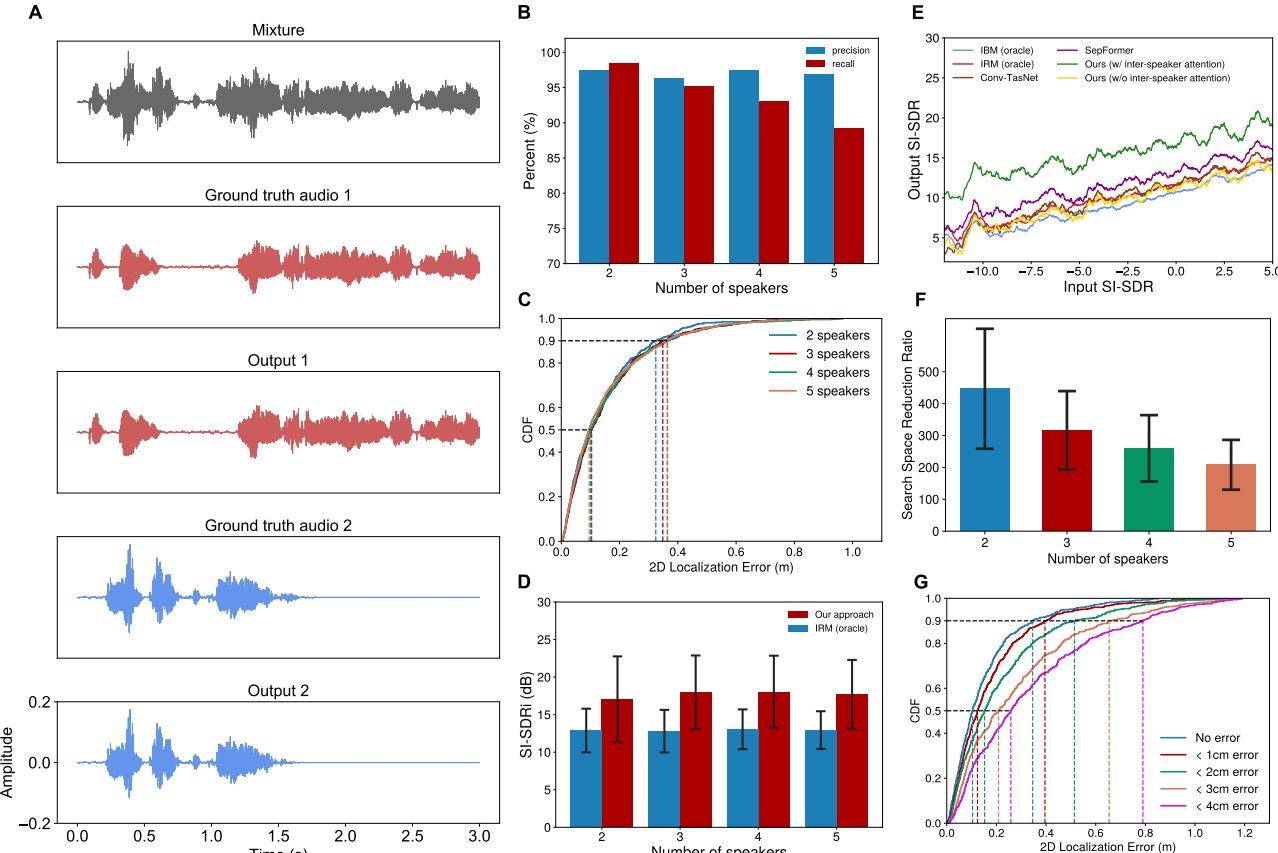

**Fig. 6 | Synthetic data evaluation. A** Shows an example time-domain mixture signal consisting of two audio signals with different amplitudes. The output of our source separation model reconstructs the two audio signals. **B** Shows the precision and recall of our system for correctly identifying the speech signals as a function of different numbers of concurrent speakers. The details of matching outputs and ground-truth are found in the supplementary materials. **C** Plots the cumulative distribution function (CDF) for the 2D localization errors for different numbers of speakers and (**D**) compares the corresponding Scale-Invariant Signal-to-Distortion Ratio improvement (SI-SDRi) with the oracle-based technique, IRM (The error bar is the standard deviation). To ensure a fair comparison, only for the oracle and neural baselines, such as IRM, we consider the top 90% of samples in terms of SI-SDRi when we report the separation results. This is because our localization stage may drop at most 10% of speakers. **E** Compares our approach with and without our attention-mechanism with oracles-based approaches and prior source separation networks (SepFormer and Conv-TasNet). **F** Shows the reduction in the search space achieved by our pruning algorithm IRM (the error bar shows the standard deviation) and (**G**) shows the 2D localization errors for different microphone position errors.

concurrent speaker cases, with the efficiency gradually decreasing to 208 for five concurrent speakers. Also, Fig. 6G and Supplementary Table 1 show that our algorithm was able to achieve a median 2D speaker localization error of 25.8 cm in the presence of the microphone position errors of 4 cm although the localization errors increased as the microphone positions became less accurate. Finally, Supplementary Table 2 shows the results for various reverberation settings, demonstrating the effectiveness of our design in the presence of multi-path interference.

We also re-trained and tested our models on a co-located circular microphone array with a 10-cm diameter and the same number of microphones. The precision drops to 71% and recall drops to 54%. This shows the importance of distributing the microphones across a larger area for 2D source localization.

Finally, we measured the total runtime of our system to process 3 s of input audio. Since the separation model used during localization is run at each location, we evaluate this model with two different sets of parameters. Both models have the same network architecture but the smaller model has fewer parameters (for detail see Methods). Figure 7A, B show that the median runtime to process a 3-s audio mixture with the smaller model was 1.82s and the 90th percentile runtime is 2.46 s, which shows our system can process the incoming data in real-time. Figure 7C shows that using a smaller separation model during localization does not cause large performance degradation.

## Real-world evaluation

We evaluated our robot swarm in real-world environments. Our evaluation used the environments that were unseen during training and included offices, living rooms, laboratories, and kitchens as shown in Supplementary Figs. 5A–C and Supplementary Fig 6. In each setting, we placed the swarm robots on a different sized surface for dispersal. To obtain the ground-truth signals, we used loudspeakers to play back speech signals from different locations in the room at heights ranging from 90 to 160 cm. Supplementary Fig. 5D shows the precision and recall for different concurrent speaker numbers. Both metrics showed results greater than 90%, demonstrating the robustness to the potential measurement errors and real-world noise and reverberation distortion. Supplementary Fig. 5E shows that the median localization error across all the tested scenarios was 15 cm and that the 90 percentile error was 49–50 cm for 3–5 concurrent speakers (Supplementary Movie 9). Finally, Supplementary Fig. 5F shows the usefulness of the inter-speaker attention, which is in line with our observations from the simulation experiments. Note that the SI-SDRi improvements were lower than those obtained for the simulated environments even for IRM using the ground-truth signals due to the considerable amount of real-world distortions.

We also evaluate the system on three surfaces with randomly placed objects that clutter the table (Fig. 8A–C). The objects are comprised of typical items found on a household table, including

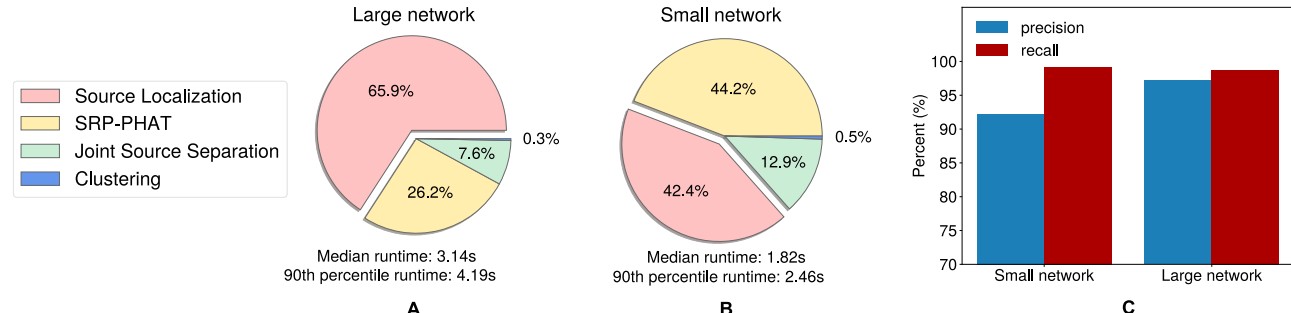

**Fig. 7 | Runtime evaluation. A, B** Shows the proportion of time spent at each step of our speaker localization and separation algorithm when using our large and small separation networks during localization. The results are for 2-speaker mixtures and for processing 3-second audio chunks. When using the larger network, the majority of the time is spent trying to localize the speakers. In contrast, when we use a smaller network, the time spent localizing speakers reduces, and the overall system runs in real time. **C** Shows that using a smaller network for localization causes only a small performance degradation.

**Fig. 8 | Real-world evaluations with cluttered surfaces and human speakers.** We assess our system's performance in three previously unseen cluttered environments, shown in (**A–C**). In (**D**), we show precision and recall results by varying the number of speakers in the audio mixture. **E** illustrates the cumulative distribution function (CDF) for the 2D localization errors across all three environments, considering varying numbers of speakers. **F** compares our approach, with and without the inter-speaker attention mechanism, and an oracle approach (IRM) (the error bar shows the standard deviation). **G** Shows the 2D localization errors for a participant in different locations. **H** Plots the mean 2D localization error as a function of the human head orientation, where 0° is when the human face is pointing in the direction of the acoustic swarm and 180° is when the human's back is facing the acoustic swarm. **I** Shows the mean 2D errors as a function of different distances of a human speaker to a wall.

phones, pens, laptops, liquid containers, booklets, boxes, and wires. These objects create occlusions between pairs of robots, as well as between individual robots and the speakers. We use our acoustic swarm localization method to determine the positions of the robots. Figure 8D shows that the precision and recall values are both above 90% and the median localization error across all these scenarios was 14 cm. Further, the 90th percentile error was 41–49 cm for 3–5 concurrent speakers (Fig. 8E). Figure 8F also shows that the separation quality is above 10 dB in the presence of clutter on the table.

We further evaluated our system with five (three male and two female) human adults who had different accents in four different rooms. The participants uttered English phrases from different

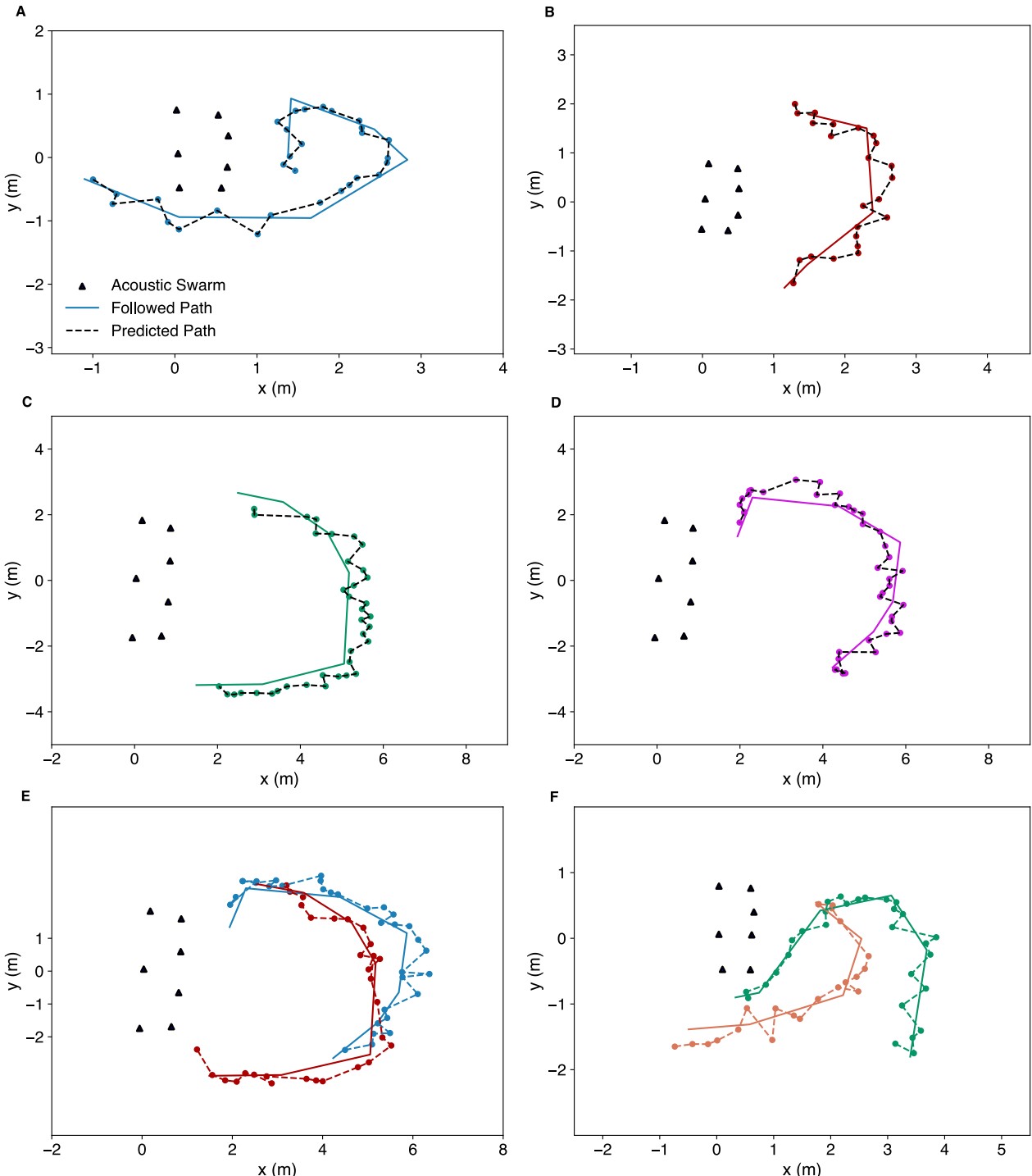

**Fig. 9 | Trajectories of mobile participants. A–D** Show the paths taped on the ground that participants were asked to follow and the trajectories predicted by our acoustic swarm of a mobile participant moving around in different rooms. **E, F** show the trajectories for two mobile participants who talked concurrently in the same room.

locations in a room. Figure 8G shows that the human speakers were detected with a median localization error of 14 cm and a 90th-percentile error of 50 cm. We also evaluated the robustness of our system to various orientations of the participants relative to the swarm. Figure 8H shows that the localization errors were low when the participant's orientation was within 135° with respect to the swarm. Figure 8I also shows that even the participants close to the walls were localized with high accuracy when their distances to the walls were larger than 60 cm.

Finally, we demonstrate various potential applications with our acoustic swarm system (Supplementary Movies 1–3). Figure 9A–D shows moving speaker tracking results, i.e., estimating the trajectory of one moving participant. Figure 9E, F show the results for two simultaneously-speaking moving talkers. In these experiments, the participants were instructed to follow trajectories marked on the floor as they spoke. All the above results demonstrate our system's ability to generalize to unseen real world environments and human speakers.

## Discussion

We presented an acoustic swarm system that can self-distribute without any cameras or external infrastructure. Using the resulting distributed microphone array, we introduced a joint speech separation and 2D localization framework that uses attention-based neural networks. Our current swarm implementation uses a single-digit number of robots as microphones, which is in the same range as those employed in commercial smart speakers such as Amazon Echo. Increasing the number of robots may provide better spatial resolution for speech separation and 2D localization. All navigation operations including time synchronization, 1D ranging, the IMU-acoustic fusion algorithm, motion planning and control, pre-entry maneuvering, and docking operations ran on-device at the robots. However, the swarm 2D localization algorithm and the joint neural network framework for speech separation and localization ran on a central base station with more computational capabilities. Finally, the localization and separation performances can be further improved by training the models on a larger amount of data collected in real-world reverberant settings.

Multiple factors affect 2D speech localization. A larger array size provides a higher spatial resolution, leading to lower 2D localization errors (Supplementary Fig. 4). This is in contrast to a commercial circular array (~10 cm diameter), where the precision and recall are reduced to 71% and 54%, highlighting the need for a distributed microphone array. Larger errors in the microphone location estimates result in larger localization errors (Fig. 6G). Human head orientation and distance to the wall can also affect the localization accuracy as shown in our real-world evaluation (Fig. 8H, I). It is noteworthy that as the number of concurrent speakers and reverberation time (RT60) increases, the median localization errors only slightly increase by around 2 cm (Fig. 6C and Supplementary Table 2), demonstrating the scalability of our system to a larger number of speakers as well as reverberant environments.

While showing a strong generalization capability to real-world reverberant environments, our system has four key limitations. First, the current navigation algorithm makes two assumptions: (1) there are no objects in the clear zone of tens of centimeters around the base station, and (2) no objects have been added later to the path taken by an external robot during dispersal. If there are obstacles in this zone, the robots may not be able to navigate back to the base station. Second, we demonstrate swarm dispersal on relatively smooth surfaces like tables. Increasing the size of the wheels and the distance between the base of the robot and the surface can enable locomotion over more uneven surfaces like carpets. Third, while speakers can be at different heights, our system only achieves 2D localization instead of 3D mainly because all our acoustic swarm robots locomote on a single 2D surface. Fourth, as our current charging mechanism is contact-based, the charging surface is susceptible to wear and tear, which can degrade charging performance over time. Since our robots are equipped with a dual-purpose AC and DC battery charger, future iterations of the robots can utilize wireless charging coils directly underneath the base station checkpoints to overcome this issue, as was done in ref. 47.

Our proposed system is an important step in the direction of achieving capabilities that have long only existed in the realm of science fiction. Our acoustic swarms present vast opportunities for novel audio applications as they can physically adapt their structures to the environment unlike the conventional centralized microphone arrays while automatically recharging on their own. For example, our swarm robots may be deployed in conference rooms to cover much wider spaces than the existing meeting devices. Our swarm also can address the long-standing cocktail party problem by allowing the user to focus on a conversation at specific regions in the room. Additionally, the swarm can be a part of future smart homes, permitting speech interaction with devices based on the speakers' locations. Finally, since our robots are also equipped with loudspeakers, future work may create distributed self-organizing speaker arrays that can program sound zones, where people in different zones of the room can perceive different sounds.

## Methods

Our research complied with the ethical regulation of the University of Washington IRB. Informed consent was obtained by participants. The authors affirm that human research participants provided informed consent for the publication of the images in Fig. 1.

### Acoustic localization

For ranging and localization, the microphones and speakers are sampled at 62.5 kHz. To perform distance measurements between robots, each robot sends a 32 ms chirp between 15 and 30 kHz and records the send timestamp, $t_{send}$. The other robots listen on their two microphones to compute the received chirp timestamp, $t_{recv}$. The robots then share their timestamps using BLE to compute the 1D relative distance as, $d = (t_{send} - t_{recv}) \cdot c + d_{offset}$, where $c$ is the speed of sound and $d_{offset}$ is the fixed distance between the speaker and microphone in the transmitting robot. In practice, accurately estimating $t_{recv}$ is challenging due to the multi-path in reverberant indoor environments. In reverberant environments, we cannot assume that the direct path has the highest power. Instead, we design a ranging algorithm that uses the two microphones on the robot to accurately estimate the direct path (see Supplementary Algorithm 1).

Our pairwise 2D localization algorithm can be abstracted as follows. We have 2 types of nodes: $N$ nodes whose positions $[\mathbf{p_1}, \mathbf{p_2}..., \mathbf{p_N}]$ need to be estimated and $M$ landmarks whose positions are known $[\mathbf{p_{N+1}}, \mathbf{p_{N+2}}..., \mathbf{p_{N+M}}]$. Say $\mathbf{D}^{(N+M)\times(N+M)}$ denotes the pairwise distance matrix, where $\mathbf{D}(i,j)$ represents the distance between nodes $i$ and $j$. The 2D localization problem for the $N$ nodes can be formulated as a minimization function,

$$\min_{\mathbf{p_1},...,\mathbf{p_N}} \sum_{1 \le i \le N} \sum_{i < j \le N} \left[ \mathbf{D}(i,j) - |\mathbf{p_i} - \mathbf{p_j}| \right]^2 + \sum_{1 \le i \le N} \sum_{N+1 \le j \le M+N} \left[ \mathbf{D}(i,j) - |\mathbf{p_i} - \mathbf{p_j}| \right]^2,$$

(1)

where $\mathbf{p_1}, ..., \mathbf{p_N} \in \mathbb{R}^2$ are the unknown 2D positions and $\mathbf{p_{N+1}}, ..., \mathbf{p_{N+M}} \in \mathbb{R}^2$ are the known 2D landmark positions. In our swarm, the $M$ landmarks are the virtual landmarks created by the motion of the robot on the platform (Fig. 2A) and the $N$ unknown nodes are the external robots to be localized. To solve this optimization problem, we use an iterative scaling by majorizing a complicated function (SMACOF) algorithm[48].

In practice, the iteration-based SMACOF algorithm may fail for two reasons: (1) inappropriate initial positions, and (2) outliers in the measured pairwise 1D distances. To address the first problem, we use tri-lateration[49] to estimate coarse positions and use them as initial values for the SMACOF algorithm. Tri-lateration uses the distances from the object to three or more known reference points to determine the object's positions. To identify outlier 1D distance measurements between the landmarks and external robots, we iteratively remove individual and pairwise subsets of 1D measurements and recompute the tri-lateration minimization function. If the optimization value reduces to less than 1% of the original value then we identify those 1D distances as outliers and eliminate them from our measurement set. To identify outlier 1D distance measurements between pairs of external robots, we compare the measured 1D distance between the robots with the distance estimated by the tri-lateration algorithm. If they differ by more than 20 cm, we identify the measurement as an outlier and remove it.

### IMU-acoustic fusion for navigation

The system makes use of two distinct reference frames: the robot's local reference frame, as defined by the axes of the IMU sensors (Supplementary Fig. 7A), and the global reference frame, as defined

by the base station (Supplementary Fig. 7B). Each robot maintains a current state vector with its 2D position, yaw (Supplementary Fig. 7C), and velocity, $(\mathbf{P_t}, \theta_t, \mathbf{V_t})$, defined relative to the global reference frame. The robots sample the accelerometer at 100 Hz and the gyroscope at 104 Hz. We apply exponential smoothing with smoothing factors of 0.9 and 0.5 to the angular velocity and acceleration measurements respectively. Suppose the accelerometer and gyroscope data at each time, $t$, are given as, $(a_t^x, a_t^y, a_t^z)$ and $(\omega_t^x, \omega_t^y, \omega_t^z)$ respectively.

The robots continuously update their state $(\mathbf{P_t^I}, \theta_t^I, \mathbf{V_t^I})$ using the IMU data as,

$$\begin{cases} \theta_t^I = \theta_{t-\Delta t}^I + \omega_t^z \Delta t \\ \mathbf{V_t^I} = \mathbf{V_{t-\Delta t}^I} + a_t^x \mathbf{u}_{\theta_t^I} \Delta t \\ \mathbf{P_t^I} = \mathbf{P_{t-\Delta t}^I} + \mathbf{V_{t-\Delta t}^I} \Delta t + \frac{1}{2} a_t^x \mathbf{u}_{\theta_t^I} \Delta t^2 \end{cases} \quad (2)$$

where $\Delta t$ is the IMU data sampling interval and $\mathbf{u}_{\theta_t^I}$ is the unit vector along the direction $\theta_t^I$. Additionally, since the robot may tilt slightly during motion, we also keep track of the robot's pitch using the gyroscope. Specifically, we track the pitch $\phi_t$ (Supplementary Fig. 7D) as, $\phi_t = \phi_{t-\Delta t} + \omega_t^y \Delta t$, where $\omega_t^y$ is the $y$-axis output of the gyroscope. We use the pitch to project the accelerometer $x$- and $z$- components to the direction of motion and use the projected values to compute state updates.

To avoid large drift errors from the IMU, we use the acoustic data to re-calibrate the current state. The moving robot sends an acoustic chirp every 200 ms, which is used by the other stationary robots to estimate its 2D position $\mathbf{P_t^A}$ using tri-lateration. To periodically fuse it with the IMU data, the robots maintain a history of $n$ positions for the moving robot inferred from the acoustic data, $[\mathbf{P_{t-n\Delta_A}^A}, \ldots, \mathbf{P_{t-\Delta_A}^A}, \mathbf{P_t^A}]$, where $\Delta_A$ is the acoustic measurement interval 200 ms. The angle and velocity can be estimated using these acoustic measurements $\mathbf{P_t^A} = [x_t^A, y_t^A]$ as,

$$\theta_t^A = \arg\min_{\theta_t^A, \epsilon} \sum_{0 \le i \le n} \left[ y_{t-i\Delta_A}^A - \tan\left(\theta_t^A\right) x_{t-i\Delta_A}^A + \epsilon \right]^2 \quad (3)$$

$$\mathbf{V_t^A} = (\mathbf{P_t^A} - \mathbf{P_{t-\Delta_A}^A})/\Delta_A \quad (4)$$

The minimization problem for $\theta_t^A$ is solved using linear regression using a bias term, $\epsilon$. Periodically, the robot can fuse the estimated state $(\mathbf{P_t^A}, \theta_t^A, \mathbf{V_t^A})$ from the acoustic data with the IMU-estimated state, $(\mathbf{P_t^I}, \theta_t^I, \mathbf{V_t^I})$. Specifically, we compute the following weighted sums.

$$\begin{cases} \hat{\theta}_t = w_\theta^I \theta_t^I + w_\theta^A \theta_t^A \\ \hat{\mathbf{V}}_t = w_V^I \mathbf{V_t^I} + w_V^A \mathbf{V_t^A} \\ \hat{\mathbf{P}}_t = w_P^I \mathbf{P_t^I} + w_P^A \mathbf{P_t^A} \end{cases} \quad (5)$$

$w_\theta^I, w_V^I, w_P^I$ and $w_\theta^A, w_V^A, w_P^A$ are the corresponding weights for the IMU and acoustic data, respectively. In our implementation, we set $w_V^I = 0.2, w_V^A = 0.8, w_P^I = 0$ and $w_P^A = 1$. If the $R^2$ coefficient for linear regression is larger than 0.8 and $n$ is greater than 4, we set $w_\theta^I = 0.2$ and $w_\theta^A = 0.8$, otherwise we set $w_\theta^I = 1$ and $w_\theta^A = 0$.

## Localization by separation

For efficiency and generalization, we operate in the uniformly spaced TDoA space[50,51]. The TDoA space is a multidimensional space where each dimension represents the time delay difference between the arrival of the signal at the first microphone and the other microphones. In the 3D Euclidean space $\mathbb{R}^3$, let $\mathbf{P_i} = (x_i, y_i, z_i)$, $i = 1, \ldots, M$ be the position of the $i$-th microphone and $\mathbf{x}$ be the candidate source position. A mapping from the position $\mathbf{x}$ in 3D Euclidean space to the position $\zeta$ in

TDoA space is defined as follows[51]:

$$\boldsymbol{\tau} : \mathbb{R}^3 \longrightarrow \mathbb{R}^{M-1}$$
$$\mathbf{x} \mapsto \boldsymbol{\zeta} = (\tau_{21}(\mathbf{x}), \tau_{31}(\mathbf{x}), \ldots, \tau_{M1}(\mathbf{x})) \quad (6)$$

Here, $\tau_{ji}(\mathbf{x}) = \frac{f_s}{c}(|\mathbf{x} - \mathbf{P_j}| - |\mathbf{x} - \mathbf{P_i}|)$ is the TDoA between the microphone $i$ and microphone $j$. We set the sampling rate $f_s = 48,000$ Hz and the speed of sound $c = 343 \text{ m s}^{-1}$.

We follow multiple steps to achieve 2D multi-speaker localization.

*Step 1. Mapping 3D space to TDOA space.* For our implementation of the mapping between 3D space and TDoA space, we use a sampling-based method. We first divide the 3D Euclidean space into smaller subspaces with dimensions $5 \times 5 \times 10$ cm. We map the centers of these 3D subspaces $\mathbf{x} \in \mathbb{R}^3$ to a point in the TDOA space $\boldsymbol{\zeta} \in \mathbb{R}^{M-1}$. We then cluster this set of points in the TDOA space into hypercubes of width 2 and 4 samples. A hypercube in the TDOA space with center $\mathbf{C}$ and width $W$ is defined as the set of points in the TDOA space whose Chebyshev distances from $\mathbf{C}$ are less than $W$. We use a dynamic programming algorithm to cluster the points into a set of hypercubes and output the hypercube centers (Supplementary Algorithm 2).

*Step 2. Pruning using SRP-PHAT.* The hypercube centers in the TDOA sample space correspond to steering vectors. Thus, we can use the sample offset of the hypercube's center to calculate its SRP-PHAT value. We apply SRP-PHAT to the width-2 hypercubes. We select the hypercubes with the SRP-PHAT values greater than both noise and those of the adjacent hypercubes as being speaker position candidates.

For each valid hypercube satisfying these conditions, we use the larger, width-4, hypercube with the same center and regard it as the source candidate. This is to account for the SRP-PHAT errors caused, for example, by room reverberation and imperfect microphone positions. Note that running SRP-PHAT directly on width-4 hypercube resulted in worse performance in our preliminary tests than running it on width-2 hypercubes and then converting them into width-4 hypercubes.

*Step 3: Source separation and clustering.* Next, we shift the microphone signals in time according to the center of each of the hypercubes resulting from the previous step and feed the shifted signal data into our separation model for localization (Supplementary Algorithm 3). The separation model outputs a signal for each hypercube. We calculate the moving average powers of these output signals with a window size of 12000 samples and a step of 1 sample. If the maximum of the moving averages is below a distance-dependent power threshold, the corresponding hypercube is regarded as not containing speakers and thus removed from further consideration, to reduce computational complexity. The distance-based threshold is used to take into account that farther sources usually have lower signal levels at the microphone positions. Then, to obtain fine-grained localization, we divide the surviving larger hypercubes, with output signal levels above the threshold, into smaller hypercubes with a width of 2. We re-apply the separation network on these smaller hypercubes and remove the hypercubes that are unlikely to be containing speakers by using the distance-based threshold (Supplementary Algorithm 5).

Finally, we run a clustering algorithm on the remaining hypercubes to remove: (a) duplicate outputs from adjacent regions in the TDOA space and (b) phantom speakers due to signal reflections aligning at other locations. The artifacts of the first type take place at locations close to the real speaker positions in the TDoA space, and the output signals of these artifacts have high similarity with those of the true positions (Supplementary Fig. 8A). To remove these artifacts, we merge small hypercubes with their neighborhood hypercubes that have the highest signal power among the neighborhoods (we define two hypercubes are neighborhoods if their Chebyshev distance $\le 4$ samples) and if they have similar output content (SI-SDR between them is greater than $-4$ dB). To calculate the speaker position for each

merged region, we find all the hypercubes within the merged region whose output signal powers are greater than 75% of the largest one and compute the weighted average of their TDoA coordinates by using the signal powers as the weights. Finally, we map the averaged TDoA coordinates back to the 3D space and use it as the predicted speaker position (Supplementary Algorithm 4). Phantom speakers caused by reverberations have multiple properties: (1) they usually occur far from the real speakers, (2) they may contain only partial segments of the original speech (Supplementary Fig. 8A), (3) they are shifted in time relative to the real speakers and (4) they may be composed of segments from different speakers in the multiple-speaker scenario (Supplementary Fig. 8B). Considering these, we first order all potential speaker outputs based on their power. We maintain a set of detected speakers, which we initialize with the single speaker with the highest power. We then consider the next highest power speaker output. To compare it with the set of already found speakers, we first split it into non-silent segments with a maximum length of 8000 samples. We then check whether these segments belong to a combination of some of the segments from the found speaker set by calculating the time-invariant SDR metric[52] across all possible combinations. If such a combination can be found (Supplementary Algorithm 6), the new speaker candidate is dropped. Otherwise, we add it to the detected speaker set. This allows us to identify the number of speakers and their 2D positions.

**Separation network used for localization.** The separation network for localization, shown in Fig. 5B, accepts the time-shifted audio signals aligned for a location of interest as input. As we use 7 swarm robots, the input is a 7-channel audio. We first apply a $1 \times 1$ 1D-convolution layer with stride 1 to increase the input channels from 7 to 64. We use a U-Net architecture with 5 encoder and decoder blocks with skip connections between the corresponding blocks. The stride length ($S$), which is the number of samples to move a convolutional kernel after each step in the convolution, is varied across the U-Net layers. The encoder and decoder blocks consist of dilated residual stacks, each containing 3 residual blocks with different dilation values. We then condition the output of these dilated residual stacks on the hypercube width parameter $W$. This is a one-hot encoding vector, which takes either [1,0] and [0,1], each representing the width of 2 or 4. Specifically, to condition the network on $W$, we use a linear transformation to the one-hot vector to obtain a different embedding vector at each block in the U-Net, and multiply it by the encoded representation of our audio signal along the channel dimension. The U-Net encoder blocks then pool the signals using a convolutional layer followed by a Gated Linear Unit (GLU). Conversely, the decoder blocks upsample the features using transposed 1D-convolutions and a GLU. The output of the last U-Net encoder layer is passed to a transformer encoder bottleneck to allow the network to attend to different temporal regions of the speech signal. This transformer encoder has 8 heads, 1024 feedforward dimensions, and is repeated twice. The U-Net output is used to compute a mask in the latent space via a 1D convolution, with a 2048 output channel, a kernel size of 33, and a stride of 16. The mask is applied to the latent representation of a reference channel (the first microphone in our implementation) by means of element-wise multiplication. We decode this masked result via a 1D transposed convolution with a kernel size of 33 and stride of 16 to estimate the clean signal that would have been observed by the reference microphone channel. The network is trained by randomly sampling points in a collection of rooms and predicting the speech signal of a target speaker at each point. If there are no speakers sufficiently close to the chosen point as specified by the hypercube width parameter $W$, then the target signal is set to zero.

### Separation by localization
Figure 5C shows the separation network used to separate individual speakers. For each of the $N$ detected speaker locations from the previous step, we align the microphone signals. The aligned signals for all the $N$ locations are then fed to the separation network. While this separation network has some commonalities with that used for localization, there are some key differences. First, the network uses a 4-layer U-Net and applies the U-Net encoder and decoder stages separately to the speaker-aligned signals of each of the $N$ speakers. Second, we use a larger feature dimension for masking (4096) and reduce the residual dilation factor to 2. The kernel size of the 1D-convolutions in the residual blocks and the encoder blocks is also changed to 5. Additionally, the bottleneck layer is used to perform self- and cross- attention using 3 pairs of conformer and transformer encoders. The conformers are applied to each speaker independently over the time dimension. The transformer encoders are then applied across speakers where the attention is on the speaker dimension. The conformer blocks have 8 heads, 1024 feedforward dimensions, and a kernel size of 31. The transformer encoders have 8 heads and 1024 feedforward dimensions. An encoded output for each speaker is produced from the bottleneck layer and passed through the U-Net decoder. For each speaker, we generate a mask and apply it to the latent representation of the common reference channel to obtain a clean speech estimate of that channel for each speaker.

## Data availability
The data used for our machine learning models have been deposited in three parts at https://zenodo.org/record/8219720, https://zenodo.org/record/8222714 and https://zenodo.org/record/8222784 under a Creative Commons Attribution 4.0 International License.

## Code availability
We provide the circuit design files used to create the robots, as well as the firmware source code at https://github.com/uw-x/AcousticSwarms-Robots. We also provide the source code for the speech processing algorithms at https://github.com/uw-x/AcousticSwarms-Speech.

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

## Acknowledgements

The authors thank Maruchi Kim, Abhishek Gupta, Vikram Iyer and Justin Chan for their feedback on the manuscript. The University of Washington researchers are funded by the Moore Inventor Fellow award #10617 (S.G.).

## Author contributions

M.I. designed and fabricated the swarm hardware. T.C. and M.I. designed and tested the navigation and localization algorithms. M.I and T.C. designed and evaluated the neural networks with guidance from T.Y. and S.G. All authors wrote the manuscript. Conceptualization: M.I., T.C., S.G.

## Competing interests

S.G. is a co-founder of Jeeva Wireless and Waveley Diagnostics. T.Y. is an employee of Microsoft. The remaining authors declare no competing interests.
