## [Peer Review File · Nature Communications]

REVIEWER COMMENTS

Reviewer #1 (Remarks to the Author):

This manuscript presents self-assembling acoustic swarms, which consist of several tiny centimeter-scale robots and a wireless microphone array. The system demonstrates the effectiveness of voice localization and speech isolation. Overall, this work shows good engineering quality, but more effort is required to prove its significance and innovation. My comments are as follows:

(1)The purposes and advantages of “self-assembling” are not very clear. How to define “self-assembling” ? How does it contribute to specific technical values (e.g., any value representing the speech isolation capability)? Is there any Figure of Merit (FoM) for this self-assembling ? And based on the FoM, can the authors present a table of comparison with other existing technology?

(2)The experimental setup for acoustic localization and speech isolation is using regularly distributed swarms on a 2D surface without objects. Did the authors evaluate the system performance with randomly placed objects on the 2D surface?

(3)Please include a discussion about the analysis of acoustic localization errors.

(4)How does the coordination of sensors (microphones, accelerators and gyroscopes) affect the localization accuracy of each robot? Please include a discussion.

(5) In the experiment, what is the time consumption of acoustic swarm dispersal and returning to the base, respectively? In addition, please detail the time consumption of separating an unknown number of concurrent human speakers and locating them. This maybe helpful to evaluate user experience.

(6) There are some small mistakes in this manuscript. For example: In page 37, the descriptions of Figure 6F and 6G do not match the contents of the two figures.

Reviewer #2 (Remarks to the Author):

This is a very good piece of work that should be published after minor corrections. The authors have done an excellent work proposing and testing a sleek algorithm for acoustic localization and source separation using a swarm of robots. The paper is generally well written and benefits from a set of supplementary material.

My main criticism is that there are abbreviations and technical jargon in general that are not defined. Surely, the authors are used to these terms, but it would be beneficial for the reader if these are properly defined and their meaning is well explained.

Also, there are a few inconsistencies in the presentation. These and other points are highlighted in the annotated copy of the manuscript that I attach to help the authors to revise their work.

Creating speech zones with self-distributing acoustic swarms

Nature Communications manuscript

We would like to thank the referees for the feedback on the manuscript. We have modified the manuscript to address the issues that were raised by the reviewers as well as to make the abstract and introduction crisper. We firmly believe that this valuable input has significantly improved our work, enhancing its potential impact.

Below we provide point-by-point responses to each of the referee comments.

Reviewer: 1

This manuscript presents self-assembling acoustic swarms, which consist of several tiny centimeter-scale robots and a wireless microphone array. The system demonstrates the effectiveness of voice localization and speech isolation. Overall, this work shows good engineering quality, but more effort is required to prove its significance and innovation. My comments are as follows:

We express our gratitude to the reviewer for their insightful comments and constructive feedback, which significantly contributed to enhancing the quality of the revised manuscript.

(1)The purposes and advantages of “self-assembling” are not very clear. How to define “self-assembling” ? How does it contribute to specific technical values (e.g., any value representing the speech isolation capability)? Is there any Figure of Merit (FoM) for this self-assembling ? And based on the FoM, can the authors present a table of comparison with other existing technology?

We thank the reviewer for raising these important points. As the term self-assembling may be a bit unclear, we instead changed the term to “self-distributing”, which is more in line with our goal of self-distributing wireless microphone arrays. A set of microphones and speakers scattered around a large area enables a wide range of applications, some of which we demonstrate in this paper. However, achieving such a setup is very challenging: 1) it is tedious and time consuming to install, 2) it is difficult to reconfigure to different environments, 3) it is hard to power and synchronize, especially as the system is scaled up. A self-distributing wireless microphone array system, capable of autonomously deploying these devices, performing some downstream task, and then retracting back when necessary, would tackle these problems, thus enabling a new suite of practical applications. This “self-distributing” property is then defined as any system that can wirelessly distribute itself across a surface without any additional setup, such as overhead cameras or projectors, required.

Once we have this self-distributing system of microphones and speakers, we can build applications that can make use of these spatially distributed devices. To demonstrate the benefits that distributed arrays have over co-located arrays, such as modern-day commercial smart speakers, we run an experiment where we replace our distributed array with a 7-element circular microphone array with a diameter of 10cm. The positions of all speakers across all samples is kept the same. We re-trained and tested our models on this co-located microphone array, and we observed that the precision drops to 71% and recall drops to 54%. This shows the importance of distributing the microphones across a larger area for 2D source localization. We added this new result to lines 341-344: “We also re-trained and tested our models on a co-located circular microphone array with a 10-cm diameter and the same number of microphones. The precision drops to 71% and recall drops to 54%. This shows the importance of distributing the

microphones across a larger area for 2D source localization.”

The specific figure of merit we use is the surface occupancy ratio. This assesses how efficiently our swarm is making use of the available space. This metric looks at the apparent microphone array sizes achieved by our swarm from different angles relative to the largest array size possible with the constraints of the given surface. A good distribution strategy is one that will maximize this ratio at all angles for different surface geometries. We show that our distribution strategy can create microphone arrays that maintain a surface occupancy ratio above 75% at all angles for circular and rectangular tables, meaning that the swarm can efficiently make use of the available space. In our updated manuscript, we have added a formal definition to this ratio (lines 911-915), as well as an accompanying illustration (Figure S11).

911 **Surface Occupancy Ratio.** Intuitively, this measures how wide an array appears at a given angle
 912 versus how wide it can appear to be for a given shape. This metric can be used to evaluate how
 913 effectively the swarms utilize the desk space. An illustration of this metric is shown in Figure S11].
 914 Formally, let p_1, p_2, \dots, p_M be the positions of robots and v_1, v_2, \dots, v_N be the vertices of the desk.
 915 Then, the surface occupancy ratio at a particular angle, θ , is formulated as follows:

$$\frac{d(\theta)}{d_{max}(\theta)} = \frac{\max_{i,j}(p_i - p_j)[\sin(\theta), -\cos(\theta)]^T}{\max_{i,j}(v_i - v_j)[\sin(\theta), -\cos(\theta)]^T}$$

Figure S11. Surface Occupancy Ratio. This metric compares the apparent array size as seen at a particular angle to the maximal array size that can be achieved with the surface used.

Fig. 3F shown above, plots the occupancy efficiency achieved by our swarm system as a function of angle for different table shapes.

Finally, to emphasize the unique aspects of our swarm system, we added a table to compare our work with prior systems with centimeter-scale robotic swarms.

Centimeter-scale swarm platform	Custom infrastructure	sub-100us time sync.	Localization range	Robot size (cm)	Edge detection
Zooids (3)	Light projector	No	meter-level	2.6 × 2.6	No
Cellulo (2 nd Rev.) (4)	Paper microdot pattern	No	–	7.5 × 7.5	No
WsBot (60)	Camera	No	meter-level	3.3 × 3.3	No
Kilobots (61)	Overhead controller	No	~10 cm	3.3 × 3.3	No
GRITSBot (62, 63)	Camera	No	meter-level	3 × 3	No
MicroMVP (2)	Camera	No	meter-level	8 × 5	No
Ours	None	Yes	~5 m	3 × 2.6	Yes

Table 1. Comparison with previous centimeter-scale swarm platforms.

As shown in the table, several previous works require an external setup for localization or communication. Additionally, they may be limited by how far they can self-distribute.

(2)The experimental setup for acoustic localization and speech isolation is using regularly distributed swarms on a 2D surface without objects. Did the authors evaluate the system performance with randomly placed objects on the 2D surface?

We thank the reviewer for this question. To address this concern, we have additionally evaluated the system in 3 distinct cluttered environments. We collect data by placing our robots on a surface with randomly placed objects and estimating their positions using the 2D localization algorithm. We then record and play back speech signals through a portable speaker at different positions in the room and mix these recorded signals to obtain 500 evaluation samples. We included images of these environments, as well as the results of this evaluation:

First environment:

Second environment:

Third environment:

Our results show that our system can still achieve a precision and recall above 90%, with median and 90-percentile errors of around 14 cm and 41-49 cm for 3-5 concurrent speakers. This means the system can work well even in cases where there are objects randomly placed on the table. We believe this is because of two things. 1) As we've shown in Fig 6G, the robots can localize themselves to within a 2cm error even in the presence of clutter, due to the redundancy created by having multiple pairwise 1D-ranging estimates from different positions across the table. 2) At any location in the room, it is highly unlikely that all robots would be occluded by objects on the table. In fact, from our observations, only 1-2 robots at most will be completely or partially blocked. This means the remaining robots may be able to compensate for these obstructions, and in some sense this is analogous to the redundancy which the robots utilize for 2D localization.

We added these results to Figure 8 and the description of these results to lines 366-373.

(3) Please include a discussion about the analysis of acoustic localization errors.

Thank you for this comment. We have now added a comprehensive discussion of speech localization errors in lines 401-409.

401 Multiple factors affect 2D speech localization. A larger array size provides a higher spatial reso-
402 lution, leading to lower 2D localization errors (Fig. S4). This is in contrast to a commercial circular
403 array (~10 cm diameter), where the precision and recall reduced to 71% and 54%, highlighting the
404 need for a distributed microphone array. Larger errors in the microphone location estimates results in
405 larger localization errors (Fig. 6G). Human head orientation and distance to the wall can also affect
406 the localization accuracy as shown in our real-world evaluation (Fig. 6H,I). It is noteworthy that as
407 the number of concurrent speakers and reverberation time (RT60) increases, the median localization
408 errors only slightly increase by around 2 cm (Fig. 6C and Table. S2), demonstrating the scalability of
409 our system to a larger number of speakers as well as reverberant environments.

(4) How does the coordination of sensors (microphones, accelerators and gyroscopes) affect the localization accuracy of each robot? Please include a discussion.

Thank you for your suggestion, we conducted more experiments on the coordinations of sensors for robot localization, navigation and control.

We added this result in lines 898-910 and Fig. S10. Specifically, we add the following to lines 898-910: "Sensor coordination among microphones, accelerators, and gyroscopes plays an important role in robot localization and navigation. To illustrate the superiority of sensor coordination, we control the robot to move in a curve. We place an overhead camera to capture the movement of the robot and use the optical flow algorithm to extract a trajectory for the robot's movement (see the blue curve in Fig. S8). Then we record the IMU and acoustic localization data and perform localization offline. In Fig. S8, the green curve is the trajectory inferred from just the IMU data (accelerators and gyroscopes). In the beginning, the path is accurate and as time goes on, the drifting becomes larger and larger. The black cross symbol is the localization result from the acoustic data and the red curve is the path from our fuse algorithm of IMU and acoustic data. As we can see, acoustic localization can help avoid the error accumulation from IMU data. On the other hand, even though acoustic localization can provide accurate localization without drifting, it can only provide some discrete points without rotation estimation. Hence, the accelerators and gyroscopes can help track the positions and orientations of robots.

As we can see, the coordination of different sensors is very important for our robot swarm. During the localization, the acoustic localization can help to avoid the drifting from IMU data, while the IMU data can be used to track the positions and orientations of robots during two acoustic measurements. In addition, during the robot control, the IMU data is also used to do the proportional-integral (PI) control. As shown in the figure below, the blue line represents the robot's forward movement without the IMU-based PI controller, while the red line depicts the robot's forward motion with the IMU-based PI controller engaged

(5) In the experiment, what is the time consumption of acoustic swarm dispersal and returning to the base, respectively? In addition, please detail the time consumption of separating an unknown number of concurrent human speakers and locating them. This may be helpful to evaluate user experience.

We thank the reviewer for these valuable questions.

We added the following text to lines 224-236 to address the first comment: “The time it takes for the swarm to disperse and return to the base is based on two key factors: 1) the size of the table, and 2) the number of robots in the swarm. We recorded the time consumption for the dispersal and return of the acoustic swarm on a 90X45cm table. For a swarm in the base station to distribute across a surface, the robots need to first identify the order in which they are placed in the base station. For a swarm of 7 robots, this took around 32 seconds. After this, the robots disperse one by one, taking approximately 11.8+-7.0 seconds per robot dispersal. In total, the sequence discovery and swarm dispersal take around 1 minute and 45 seconds. To return to the base, the robots first moved towards the clear zone in less than 1.5 seconds. Then, the robot at the base station goes around the landmarks on the base, performing acoustic chirps along the way. This is dependent on the number of landmarks on the base station, and with 7 landmarks, it took around 30 seconds. Next, the robots outside the base take turns emitting chirps, which took about 3.1+-0.02 seconds per robot. Finally, each robot needs to navigate to the base station one by one, perform the pre-entry calibration maneuver, and successfully dock with the station. This took 40.4+-4.5 seconds per robot.”

To address the second comment, we performed further optimizations on our network to create a small network that can run in real-time. We added these results in lines 344-350 and Fig 7. Specifically, we measured the total runtime of our system to process 3 seconds of input audio. We ran our pipeline on a GPU machine (details in methods). Our pipeline can be decomposed into 4 steps: SRP-PHAT, source localization, clustering, and joint source separation. Since the separation model used during localization is run for each location, we evaluate this model with two different sets of parameters. Both the models have the same network architecture but the smaller model has fewer parameter numbers (for detail sees Methods). Figs. 7A,B show that the median runtime with the smaller model was 1.82s and the 90-th percentile runtime is 2.46s for running a 3s audio segment, which shows our system can process the incoming data in real-time. Fig. 7C shows that using a smaller separation model during localization does not cause a large performance degradation.”

We also add details about the small network in the supplementary material.

(6) There are some small mistakes in this manuscript. For example: In page 37, the descriptions of Figure 6F and 6G do not match the contents of the two figures.

We thank the reviewer for their attention to detail. We have fixed the captions to better reflect the contents of the figure.

Reviewer: 2

This is a very good piece of work that should be published after minor corrections. The authors have done an excellent work proposing and testing a sleek algorithm for acoustic localization and source separation using a swarm of robots. The paper is generally well written and benefits from a set of supplementary material.

We thank the reviewer for their positive comment.

My main criticism is that there are abbreviations and technical jargon in general that are not defined. Surely, the authors are used to these terms, but it would be beneficial for the reader if these are properly defined and their meaning is well explained.

Also, there are a few inconsistencies in the presentation. These and other points are highlighted in the annotated copy of the manuscript that I attach to help the authors to revise their work.

We thank the reviewer for their time spent reviewing the manuscript. We greatly appreciate their support and valuable remarks.

1) Page 4, referring to citations, for example “(1–14)”: It is not a good idea to use block-citing. The authors should either describe the relevance of each of these papers to their work or cite key original contributions to this area of research.

We thank the reviewer for the suggestion. We have avoided using block citations and have instead only cited specific papers when we discuss their relevance to our work.

2) Page 6, referring to “acoustic chirps”: Define the chirp parameters used. Frequency range, modulation method and duration.

We appreciate the reviewer’s suggestion. During localization, the robots emit frequency modulated continuous wave chirps between 15 and 30 kHz, each with a duration of 32 ms. These details are found in the Methods section, and to better guide readers to this information, we have added these details to line 437-438.

3) Page 6, referring to 62.5kHz: Why 62.5 kHz if the sampling rate is 48 kHz?

We thank the reviewer for the question. The robots use their microphones to record acoustic chirps as well as record human speech for downstream applications. During localization, the robots record chirps at 62.5kHz, since a higher sampling rate lets us use a larger bandwidth for the chirps and obtain finer 1D ranging estimates. However, when running our speech algorithms, the robots record speech at 48kHz, since this is the largest sampling rate our audio codec supports. When reporting our synchronization results, our chirps are sampled at 62.5kHz. Furthermore, note that the 62.5kHz chirp recordings are short in time (50ms), so the robots store them in their memory and process them on device to compute the time-of-flight, without having to stream any recorded chirp audio or processing with codec.

We have made multiple modifications to the paper which would hopefully make this clearer:

- 1) We indicated that the acoustic chirps are sampled at 62.5kHz in the previous paragraph, specifically at lines 436-438.
- 2) We added a statement to line 393-395 to clarify that the distance estimation algorithm runs on the robots

4) Page 7, referring to “(1)”: be consistent with number lists. Sometimes you use x) other times (x)

We thank the reviewer for raising this remark. We have modified the paper to stick with the notation x) consistently in the entire paper.

5) Page 9, referring to “surface occupancy ratio”: Define this ratio

We thank the reviewer for the suggestion. We have added the following text to lines 911-916, as well as an accompanying illustration (Figure S11). Intuitively, this measures how wide an array appears at a given angle versus how wide it can appear to be for a given shape. This metric can be used to evaluate how effectively the swarms utilize the desk space. We have included an additional figure illustrating the physical interpretation of this ratio:

Let $\{p_1, p_2, \dots, p_M\}$ be positions of robots and $\{v_1, v_2, \dots, v_N\}$ is the vertices of the desk.

Then the occupancy ratio at a certain angle θ is formulated as follows:

$$Ratio(\theta) = \frac{d(\theta)}{d_{max}(\theta)} = \frac{\max_{i,j} (p_i - p_j) [\sin\theta, -\cos\theta]^T}{\max_{i,j} (v_i - v_j) [\sin\theta, -\cos\theta]^T}$$

Figure S11. Surface Occupancy Ratio. This metric compares the apparent array size as seen at a particular angle to the maximal array size that can be achieved with the surface used.

6) Page 9, referring to “periodic chirps”: At what repetition rate?

Thank you for this question. When the robots move back towards the clear zone, the robot inside the

base station will emit a chirp every 200ms, i.e. at a rate of 5Hz. We have clarified this in line 196 and 482-486.

9) Page 11, referring to “SRP-PHAT”: Decipher this and/or make reference to the paper where this algorithm is properly described. It makes sense also to explain what does this algorithm usually does,

We thank the reviewer for raising this suggestion. We have edited lines 266-272 to address this concern: ” To efficiently search for speakers in the 3D space, we combine neural speech separation and a conventional source localization method. Specifically, we first prune the search space using the Steered-Response Power Phase Transform (SRP-PHAT) algorithm (48) (Fig. 5A). SRP-PHAT is a signal processing technique that can achieve coarse localization of the sound source by analyzing the phase differences between all pairs of microphones. SRP-PHAT outputs the power of the signals aligned to each possible candidate point in the search space. We prune the search space by discarding the region with low output power.” We have included this exact description in our manuscript.

10) Page 11, referring to “L1”: What is L1? Define

The L1-norm between two vectors is the sum of element-wise absolute differences between the two. We use the L1-norm between the target signal and the model output as the loss function to train the separation network for localization. When we use the L1-norm as a loss function, we refer to it as the “L1 loss”. We have included an explanation of this in our updated manuscript in line 1013 and 1025-1026.

11) Page 12, same as (4)

Please see our answer to (4)

12) Page 12, referring to “one-hot”: What is that?

We thank the reviewer for the question. In general, a one-hot encoding is a technique used in deep learning to select one setting out of a set of various distinct options. In the case of this work, we use the one-hot vector to choose between extracting speech from large regions (to eliminate regions faster) and extracting from small regions (to obtain a finer resolution). In practice, this is a vector with all elements equal to zero, except for one element, which is set to one. The placement of this one indicates the choice of setting. In our case, we have two possible settings, we either want the network to consider large regions or to consider small ones. So, the one-hot encoding will be a vector of 2 elements, one of which will be zero, and the other will be one. In the manuscript, we added a sentence explaining this in line 293-294: “This is a 2-element vector with one entry set to zero and the other set to one, depending on whether we are conditioning on small regions or large regions.”

13) Page 12, referring to “perform the search”: search for what and how?

We rewrite the sentences from the line 296 to 299.

The search means speakers search. The search process is that we apply the neural network at the candidates positions and check its separated signal (see line 264-282).

14) Page 12, referring to “spaced 3D regions”: 3D for what? It is confusing since you have been

referring to a 2D problem before.

We appreciate the reviewer's comment. Our 2D speaker localization pipeline is a search-based method. While our objective is to perform speaker 2D localization, we conduct the search for speakers in the 3D space. This is because the height difference between speakers and microphones introduces additional time delays to the multi-channel signals. Hence, without the consideration of the height difference, the signal may not be aligned even at the correct 2D position. To address this we have incorporated additional explanations to lines 264-266.

15) Page 12, referring to "cross-attention": What is 'cross-attention'? Define. 16) Page 13, referring to "self-attention": Define it

We appreciate the reviewer's comments. After some consideration, we have decided to instead use "intra-speaker attention" and "inter-speaker attention".

We have defined these terms in lines 312-314 ("intra-speaker attention") and 314-316. ("inter-speaker attention")

17) Page 13, referring to the figure with two-speaker separation waveforms: It makes sense to explain what were characteristic of these speak realizations. How long and broad in terms of frequency composition.

We thank the reviewer for the thoughtful suggestion. To illustrate the frequency composition of the two speaker signals, as well as the network output, we added a spectrogram representation (Fig. S3) of all the signals we showed in Figure 6. Notice that the two speakers overlap in the frequency domain. Additionally, they also overlap in the time duration as shown in Figure 6. We think this example showcases that our network is in fact capable of separating speakers that overlap in both time and frequency.

Figure S3: The spectrograms corresponding to the waveforms shown in Figure 6A. The mixture signal is shown in A). We show the ground truth audio for the first speaker in B) and the corresponding network output audio in C). Likewise, the spectrograms for the ground truth audio of the second speaker is shown in D) and the network output for this speaker is shown in E).

18) Page 13, referring to "ideal ratio mask": explain it or make reference to a paper explaining it.

We thank the reviewer for suggesting this improvement. To address your comment, we have added a citation to a paper which discusses IRM in line 326.

19) Page 13, referring to “SI-SDR”: what is that? Define

We have now cited a paper that defines the SI-SDR metric in line 324-325. Thank you.

20) Page 14, referring to “included offices, living rooms, laboratories, kitchens, and classrooms”: Provide an estimate for the size of these rooms and expected reverberation times.

We thank the reviewer for the question. Please find below a table that summarizes the details requested for each room. To provide the expected reverberation times, we report the RT60, which is the time it takes for a sound to attenuate by 60dB. We measure this value by placing our robots in the same configuration as we did during our recordings, and begin streaming 10 seconds of audio from all robots. During these 10 seconds, we pop a balloon to obtain an approximate room impulse response at every robot. Then, we find the peak value of each channel and consider the impulse response to start at this sample. Since the rooms we measure are noisy, we use the PyRoomAcoustics `measure_rt60` to first measure the RT20 (time to decay by 20dB), and we extrapolate to obtain an estimate for RT60. We use all robots to take 7 different measurements of the RT60 and report the average and standard deviation. This table was added to the supplementary material as Table S3.

Room	Purpose	Dataset	Size (meters)	RT60 (s)
1	Conference Room	Train	9x6x3	0.69+/-0.04
2	Office	Train	6x6x3	0.42+/-0.04
3	Classroom	Train	9x9x3	0.50+/-0.02
4	Laboratory	Train	10x5x5	0.78+/-0.07
5	Office	Validation	10x5x3	0.47+/-0.02
6	Living Room	Test	5x5x6	0.72+/-0.03
7	Laboratory	Test	10x10x3	0.41+/-0.01
8	Kitchen	Test	5x5x3	0.49+/-0.04
9	Office	Test	10x5x3	0.46+/-0.02
10	Office with Living Area	Test	9x9x3	0.50+/-0.05

Table S3: Description of rooms used for real world evaluation. Rooms used in our different dataset splits, as well as their corresponding sizes and approximate reverberation times.

21) Page 16, referring to 62.5kHz chirps: Earlier on you stated 48 kHz. Which one is correct?

Please see our response to 3)

22) Page 16, referring to chirp parameters: Suggest to state this earlier on in the paper when you talk about the experimental setup or refer to this section.

Please see our response to 3)

23) Page 17, referring to “ p_1, \dots, p_N are the unknown 2D positions”: should p_i be a 2-component vector with x and y coordinates?

That is correct. These are 2D-component vectors, the positions of the robots on the surface. We have added an indication in lines 451-452 that these are vectors in \mathbb{R}^2 .

24) Page 18, referring to “tri-lateration”: define this term.

We thank the reviewer for requesting this clarification. Tri-lateration is a traditional ranging-based localization algorithm that uses the distances from the object to three or more known reference points to determine the object’s positions. We added extra explanation and reference in lines 459--460.

25) Page 17, referring to the mean filter window size used by the gyroscope: “window size of 2”

We thank the reviewer very much for this remark. The phrase we use is inaccurate. We applied exponential smoothing with a smoothing factor of 0.9 for the gyroscope measurements, and a smoothing factor of 0.5 for the accelerometer measurements. We have modified the paper to reflect this in lines 472-474

26) Page 18, referring to the variable t, x, y, z: Italic

We thank the reviewer for their attention to detail. This has been fixed in line 474 and 480

27) Page 18, referring to the “pitch ϕ_t ”: Is the pitch the same as the orientation?

We thank the reviewer for requesting this clarification. We have added Fig.S7 to make this more clear. We define the robot axes as in Figure. S7A, and it moves in the forward direction along the x-axis in its own reference frame, as in Figure. S7B. The global reference frame is defined according to the base station, as shown in Figure S7C. The orientation is the rotation of the robot’s local XY-plane about the global XY-plane. We compute it by accumulating yaw rate measurements from the gyroscope over time, and add it to the initial orientation (90 degrees at the base station exit). On the other hand, the pitch is the rotation about the robot’s y-axis (Figure S7D). During navigation, a robot may tilt, for example, due to robot wobbling/teetering, some irregularity on the surface, or a ramp on the base station. When it tilts, the gravitational acceleration vector may shift slightly, and it may have a planar acceleration component. So, it is important to correctly sense this tilt using another method to remove the effect of gravity before updating the robot position due to linear acceleration. So, we use the gyroscope, which is insensitive to the gravitational acceleration, to keep track of the pitch. While it is possible and more complete to also implement roll tracking as well, we observed that the system functioned well enough without one.

Figure S7: Robot local reference frame and swarm global reference frame. A) A robot's XYZ-axes according to its own reference frame. B) The global reference frame as defined according to the base station. C) The definition of a robot's orientation with respect to the axes of a translated global reference, centered at the robot. D) The robot's pitch as defined relative to gravity.

28) Page 19, referring to the section on Mapping 3D space to TDoA space: This needs better explaining. Where does the 3rd dimension comes from? Height? Angle? What is TDOA space?

We now define TDoA in lines 257-259, define the 3D space in lines 494-495, and define the TDOA space in lines 494-495.

The TDoA (Time Difference of Arrival) values are the signal propagation time differences from the candidate location to each pair of microphones.

The 3D space is the normal Euclidean space including x, y, and z.

The TDoA space is a multidimensional space where each dimension represents the time delay difference between the arrival of the signal of the first microphone and the signals of other microphones. We also add the citation of TDoA in the paper.

29) Page 19, referring to “ $f_s = 48,000 \text{ Hz}$ ”: Again, you switching liberally between 48 and 62.5 kHz. Which one is correct?

Please see our response to 3)

30) Page 20, referring to the term “TDoA” multiple times: See 28)

Please see our response to 28)

31) Page 20, referring to the enumeration scheme “(a)”: You switch liberally between (letter) and (number). Suggest you stick to one style and use it throughout.

We thank the reviewer for the suggestion. We modified the enumeration style to only use numbers throughout the paper.

32) Page 21, referring to “stride”: What is that? Define.

We appreciate the reviewer’s recommendation. In a convolutional neural network, The stride length (S) is the number of samples to move a convolutional kernel by after each step. In the manuscript, we have added it in line 557-558.

33) Page 22, referring to “3e-4”: is this 3×10^{-4} ? Suggest to use a proper maths.

We thank the reviewer for the comment. We have addressed this issue by using the suggested notation. Now the revised text can be found in line 1033, 1035, and 1039.

REVIEWERS' COMMENTS

Reviewer #2 (Remarks to the Author):

The authors responded well to my previous set of comments. I am happy to recommend the publication of this revised paper in its current state.

Creating speech zones with self-distributing acoustic swarms

Nature Communications manuscript

Reviewer: 2

The authors responded well to my previous set of comments. I am happy to recommend the publication of this revised paper in its current state.s:

We express our gratitude to the reviewers for their feedback that significantly improved our manuscript.
Thank you.